# A recipe for systems change: Predictive modeling and street-level bureaucracy among homeless services

**Curtis Smith**[ID][1]*, **Moinak Bhaduri**[ID][2]

1 Sociology Department, Bentley University, Boston, Massachusetts, United States of America,
2 Mathematical Sciences Department, Bentley University, Boston, Massachusetts, United States of America

* csmith@bentley.edu

## Abstract

*This study analyzes the necessary components to managing a successful systems change by considering the processes of housing people experiencing homelessness across a range of geographic locations in the United States. Utilizing Michael Lipsky's notion of street-level bureaucracy, a nonprofit organization called RE!NSTITUTE™ challenges social service agencies to a "100-Day Challenge™" by granting front-line workers ownership of improving their work free from oversight from their administrators, which results in improved outcomes. This paper also gives legitimacy to existing models on systems change by using novel mathematical predictive modeling through using principles of conditional probability to recognize the best "recipe" for systems change. This quantitative approach achieves three key goals. Firstly, it confirms the existence of a hierarchy that suspects transformative change is the most crucial to bring about, followed by relational or mental changes, and imparts a data-driven concreteness to the theoretical model, revealing varying shares within each broad level. We innovate alternate quasi-periodic visuals whose impacts are lasting. Next, it offers (even under partial knowledge of the predictors) correlational guarantees that one may use while designing future studies: when changes in relationship, practice, and power dynamics are sure, for instance, changes in mental structures often follow with 50% chance. Finally, we point out, through change-point analysis, how shifts in the rate of reducing homelessness may be attributed to assignable causes such as understaffing, which can hamper successes and workers' ability to maintain consistent attendance. We quantify the impact of those shifts. Several instances of 100-Day Challenges™ across various regions of the U.S. are analyzed to unearth commonalities. Thefindings we offer stress that while variables such as changes in resources (or funding) are often emphasized to achieve social change, changes in practices and relationships (networking) are the most influential ingredients to achieve transformative systems change. The sustainment of workers in their positions is a key*

**Data availability statement:** The data can be found on Figshare. DOI: https://doi.org/10.6084/m9.figshare.29286521.v1

**Funding:** MB has received funding from the American Mathematical Society and the Simons Foundation. MB and CS have received funding from Bentley University.

**Competing interests:** NO

*driver in maintaining growth in successfully relocating individuals experiencing homelessness.*

## Introduction

Homeless intervention has long been approached through strategic enactment of social policies meant to inform and direct social service work [1–3]. However, many front-line workers may not accept and follow through with the implementation of enacted policies, perhaps due to flaws in such plans, whether perceived or legitimate [4]. While some administrators of public policy might focus on dictating buy-in from workers through punitive means, it is important to recognize that more effective implementation approaches may be available, and ground-level employees often have insight into these alternatives.

Workers often implement their own strategies that are different from those mandated by their administrators for various reasons. They may be more in touch than administrators with the needs of the populations with which they work, be more aware of limitations in the referral process and the idiosyncrasies of other workers with whom they interact, or be better at managing efficiency pertaining to the demands and the limited resources of their job [4–7]. Informed by literature on systems change and Michael Lipsky's [5] notion of street-level bureaucracy, we analyze data from a company called RE!NSTITUTE™ and their 100-Day Challenge to recommend operational tweaks so that people experiencing homelessness can be better served.

### Systems change: A brief synopsis

Theories on the concept of systems change date back to the 19th century, spanning several academic fields, and are perhaps best exemplified by the work of the sociologist Emile Durkheim [8]. While recent efforts have developed to measure systems change among nonprofit work, efforts to solidify results from hard data remain elusive and largely temporal. Foster-Fishman and Behrens [9] define systems change as "an intentional process designed to alter the status quo by shifting and realigning the form and function of a targeted system" (p197). However, Coffman [10] says regarding applications of the concept:

> In practice they have been more a way of describing system elements and systems initiative complexities than an evaluation methodology that spells out initiative assumptions and ways of testing whether they are valid" (p1).

Organizations such as the Nicholson Foundation have also engaged in measuring systems change for decades, underscoring the importance of engaging with government, applying best practices, nurturing partnerships, investing in organizational infrastructure, addressing complex issues with diverse strategies, and developing future leaders [11]. These efforts supplement a rising strong brand of increased attention, such as Coffman's [10] "call to action" (p1), but more work is needed. Several interpretations and strategies have evolved to measure systems change among

organizations. Brown and Rosser [12] emphasize utilizing Results-Based Accountability and System Expertise and Leadership, in building partnerships and collaboration to guide strategic decisions in grantmaking. Although there is an abundance of research on improvements to social policies, a significant gap remains among *applied* research that focuses on the implementation of such policy as it relates to systems change, and their quantitative legitimacy.

Kania et al. [13] has since utilized literature on systems change to generate key necessary components for strategic implementation. In their model, the conditions that hold systemic problems in place are identified as being related to: policies, practice, resource flows, relationships and connections, power dynamics, and mental models. They situate these themes in the following chart.

The model shown in Fig 1 demonstrates that to achieve transformative change requires many components. A change in a system depends not only on structural elements but also on how relationships and cognitive frameworks function together. At the top of this hierarchy are "structural change" aspects such as policies, practices, and allocation of resources. These items are straightforward to identify and modify, but the focus often shifts to more "relational change" in the middle row of the hierarchy. Items in the "relational change" row involve understanding people's roles, how they interact with each other, and how these interactions keep the system running. Changing these relational aspects is more complex than "structural change." However, making relational changes can lead to significant and lasting improvements in the system. Items in the "relational change" row recognizes that the way relationships and power structures function directly affect how policies, practices, and resources are managed, ultimately influencing the system's overall performance.

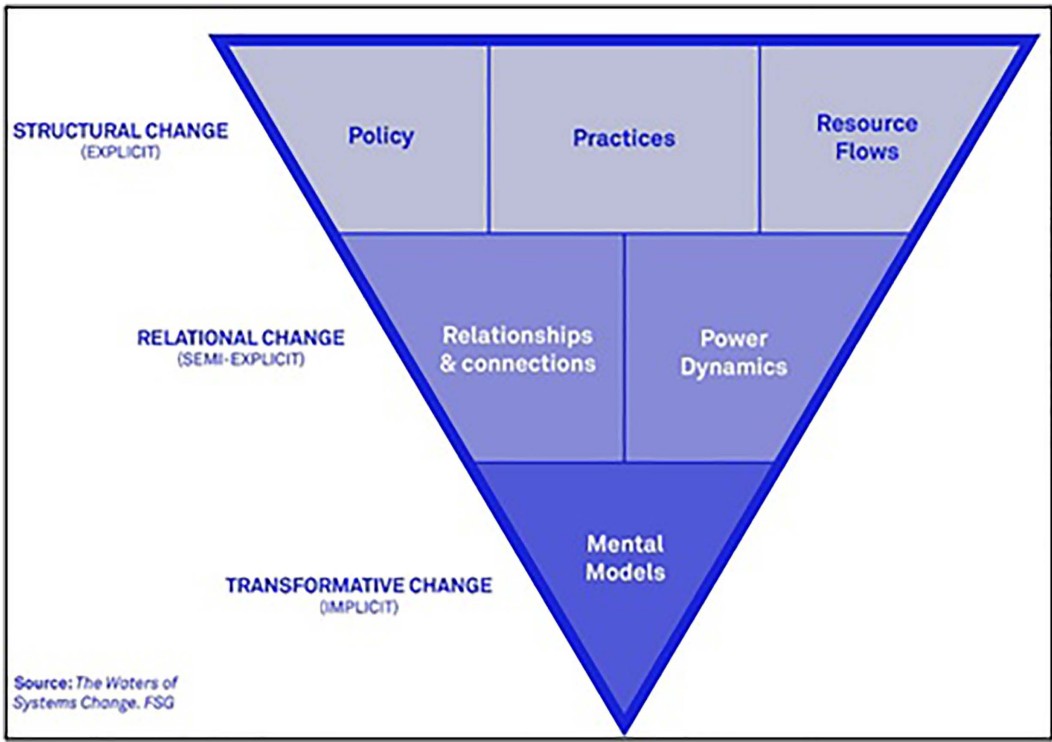

**Fig 1. Theoretical contributions from Kania et al. (2018).** A conceptual triangle is hypothesized which brings out educated guesses on how common certain kinds of changes are likely to occur in relocation exercises. The wider the base, the easier to bring about that change. We notice, therefore, how structural changes are guessed to be easier to bring about than mental changes. We notice, however, two points. First, in this conceptual map, there is no backing from observed data. Second, the cosmetics do not reveal insights: that is, the base or the altitude of the component shapes do not hold special significance. Both these points are taken up in Fig 2, a realized version of this conceptual triangle through statistical metrics.

The end point (bottom) of the hierarchy, is where "transformative change" happens, which is when mental models, i.e., people's mindsets towards issues, are changed. Changes in mental models are the most difficult to bring about because it includes how humans make sense of the world around them. Kania et al. [13] says, "Our mental models shape the meaning we assign to external data and events and guide our participation in public discourse" (p8-9). While structures can be changed and relationships can change over time, transformative change is often prolonged due to the necessity to change internal beliefs. However, once mental changes are achieved, real and foundational (i.e., transformative changes) can result. Simply put, changing mental models accelerates and promotes changes throughout the whole system, impacting all components in the pyramid. For this reason, transformative change and changes in mental models are the most desirable to achieve among systems change research.

Kania et al. [13] and the concepts in Fig 1 have been utilized to apply theories on systems change to assist in a more actionable manner among student enrolment, philanthropy, and services for children and families [10,14,15];. For example, Wallace and York [15] measured perceptions of equity among students finding a higher desirability of implicit change (i.e., relationships and power dynamics), and a majority of concerns with resource flows, which they go on to attribute to mentorship issues in their discussion and point back to power dynamics. We examine these exact operational indicators and the dependencies among them in our analysis sections below. Greenfield and Pope [16] utilize Kania et al's [13] work to analyze how Age-Friendly Community (AFC) initiatives impact systemic conditions. Focusing on eight initiatives in northern New Jersey, they find strong evidence that their initiatives improved practices, resource flows, relationships, and emphasize the importance of leadership roles but that worker turnover can hinder the important role of leaders [16]. Through offering a sound framework with which to understand the connections and dependencies among several moving pieces that go inside a successful relocation experiment that Greenfield and Pope [16] avoids we, through this study, impart stronger mathematical rationale, that are implementable along with interactive dashboards that make the connections apparent to practitioners.

## Michael Lipsky's "street-level bureaucracy"

In his seminal book, *Street-Level Bureaucracy: Dilemmas of the Individual in Public Service*, Michael Lipsky's [5,6] concept of "street-level bureaucracy" spotlights how ground-level workers creatively negotiate the demands of their employment. According to Lipsky [6], street-level bureaucrats are any community service workers who interact with the public directly and routinely, such as teachers, police officers, and social service workers [6]. Lipsky argues that despite being at the bottom level of their employment hierarchy, street-level bureaucrats often exercise substantial discretionary power by how they interpret and carry out the policies in their day-to-day work routines, especially considering that the resource allocations at their disposal are often extremely limited [6]. Examples of this could be a teacher who caters to the needs of students that they like versus other students or a police officer who plays all three branches of government (the roles of judge, jury, and executioner) as they shoot their guns at criminals on the streets.

Many researchers use Lipsky's theoretical framework to demonstrate how workers in fields such as social work, teaching, and law enforcement fail to meet policy expectations by prioritizing their own judgment over official job standards [17,18]. A substantial portion of this research is devoted to their interactions with clientele according to the race [19–21]. For example, Masters, Lindhorst, and Meyers, [22] use Lipsky's work in their study entitled *Jezebel at the welfare office: How racialized stereotypes of poor women's reproductive decisions and relationships shape policy implementation* to show that social service workers in three states used language that negatively reflected myths centered around sexuality and motherhood abilities with their Black clients, ultimately denying them services [22]. Cronley [23] found that social service workers have racial biases favoring White female clients at the expense of Black populations when administering the requisite assessment tools for service eligibility and prioritization. In a sociological review of disciplinary welfare provision, Schram et al [18] found that workers significantly used their discretion to punish Black populations, labeling them "noncompliant" and sanctioning them as they sought service aid. Prior sanctions were significant for Black populations in

denying them access to resources but seemed irrelevant for White populations [22]. These findings align with a long line of research on structural racism in the United States [24] and the relationship between race and the welfare state [25,26].

However, workers also have the ability to outperform their official duties according to Lipsky [5,6], which is a focus of this study. For example, Brodkin [27] says about Lipsky's work:

> He challenged analysts, deeply accustomed to thinking hierarchically about bureaucracy and focused on "gaps" and "compliance," to move beyond these conventional modes of thinking. His novel approach launched a still-ongoing debate about so-called bottom-up and top-down views of policy implementation. (p941)

In our own work, we [4] agree with this side of Lipsky's theories on Street-Level Bureaucracy, finding that social service workers often use their creative discretionary power to overcome what they believe to be short-sighted policy requirements. They go beyond the official requirements of their job to use "assertive advocacy", often working longer workdays or implementing nontraditional workarounds to ensure that their homeless clients' needs are met [4].

In this regard, workers are "virtual policy makers" as they carry out the demands of their employment [6,28]. While this creative discretion allows street-level bureaucrats to discriminate or fall short of policy initiatives, it also allows ground-level workers to adjust their daily routines to *better* meet the specific needs of their clientele and to satisfy difficult tasks of their job, expediting their work with the people they are meant to serve. It also means that constraining such discretion through punitive oversight can inhibit the success of ground-level workers in satisfying the goals of their employment [6]. This means that removing such oversight could improve service outcome. Drawing from Lipsky's [6] framework, we can analyze the effectiveness of street-level bureaucrats within various homeless housing systems and their institutional settings. By examining data from ground-level workers, we can derive insights regarding the impact of administrative oversight on their ability to perform their duties. We provide evidence that by removing bureaucratic red tape, we can: (1) confirm the validity of existing theoretical perspectives on the subject of systems change by providing new quantitative evidence and scaffolding (based on a structure that queries the presence of a hierarchy about how interactions and work are perceived and prioritized); (2) predict systemic change to formulate strategies for systems change, even under incomplete knowledge of the predictors; (3) quantify the impact of sudden shifts in how business is conducted, identifying a value for workers' presence (i.e., how much success is sacrificed without their presence). Additionally, we ask: Is there an overreliance on administrative oversight to implement new policy and provide more resources to workers, and conversely, could such perspectives stifle meaningful systems change?

## RE!NSTITUTE™

Founded in 2007, RE!NSTITUTE™ is an international nonprofit organization operating in such geographic locations as Europe, the Middle East, Sub-Saharan Africa, Latin America, and the United States, addressing various issues such as homelessness, domestic violence, healthcare, governance, and criminal justice [29]. One of RE!NSTITUTE™'s key methodologies is their application of Lipsky's [6] work in conjunction with their 100-Day Challenge™. When RE!NSTITUTE™ convinces agencies to grant discretionary power to their lower-level employees for 100 days, service goals often improve, work is streamlined, and the quality of services improve. To achieve this, RE!NSTITUTE™'s 100-day challenge™ emphasizes shifts in decision-making powers to spotlight the voice of frontline staff, which often results in recognizing the lived experience of their homeless clients, promoting leadership that is more open to innovation, and improving overall understanding between various roles among the larger social work system.

However, details of RE!NSTITUTE™'s success are unclear. What are the drivers for the range among their improved outcomes. Also, how can they refine a strategy that will maximize the impact of their work?

In an effort to understand the systemic transformations that they observe, RE!NSTITUTE™ began charting the emerging innovations as potential building blocks of systems identified in The Waters of Systems Change model by Foundation Strategy Group (FSG) [13].

RE!NSTITUTE™'s 100-day Challenge™ utilizes Lipsky's notion of "street-level bureaucracy" to implement systems change. Until now, the alignment of how policies, practices, resource flows, relationships, power dynamics, and mental buy-in culminates into a transformative change has been available only through a theoretical model (see Fig 1), which has not been tested empirically. Here, we design a "recipe" to see what we can expect when certain parts of the model are present over others, such as a change in resources versus a lack in relationships and connections. We test what exists in our data by showing what should be expected if we have some ingredients but not others, and spotlight which parts are more important for systemic change.

## Concrete manifestations of systems change

The model in Fig 1 portrays a summation of prominent theories on the concept of systems change [13]. This theoretical model provides a framework for understanding the critical components needed to manage systems change effectively. The items within the model have previously remained abstract, signifying an intellectual focus, so the model lacks priority and emphasis in a tangible way. We contribute actionable specifics related, but not limited to: how frequently or significantly the components should be represented, whether the constituents of these items should enjoy equal share and priority among the larger model. A key purpose of this section is to settle these technicalities. We clearly delineate where one item ends and the next begins, in actual terms that go beyond such theoretical approximations shown in Fig 1. We strive to provide the theoretical structure with a concrete foundation that is data driven. That desire is not purely academic. Unless we witness a realized version of the theoretical triangle, we cannot answer predictive questions, such as assuming that some specific changes are brought about or what other changes are likely to follow. Our grander goal is to check whether empirical data suggest the existence of this triangle model (Fig 1) at all. We test our data to find support or refute the theoretical shape. A subtler task is to check for stability and replicability: if someone else collected similar data, how much would their triangle model differ from ours?

In mathematical terms, we consider the notions originating from a field of data science called "market basket analysis" [30] deployed primarily to work out what items in a shopping context would go with other like items. Bhaduri (2023) offers a brief introduction to the topic showing how asymmetries exist in a buying-and-selling context. How, for instance, when one buys a high-value item (such as a house), certain lower-value items (such as furniture) are frequently bought (this is in keeping with the spirit of recommendation we are going to explore in the next section) but this connection does not flow the other way. Expensive items – or those purchased less frequently and inexpensive items – those purchased more frequently – may be described through the idea of varying *supports* – a notion that the next section elaborates. For instance, if we survey items 100 people bought, we may observe furniture purchased 90 times while a house purchased only once. The support for furniture would then be 0.9 (=90/100) while for house, 0.01. Were we to construct a triangle like Fig 2., the base of "furniture", therefore, would be 90 times wider than that of "house".

## Predicting support: Two metaphors

The use of Bhaduri's (2023) mathematical concepts can be seen in the following two metaphors to base some predictions. If we are shopping at a grocery store and see what is in someone else's cart, then we may be able to predict what other items they may buy next due to the items' similarities. If we notice cough medicine and a thermometer, then we may predict that they will buy hot soup or throat lozenges next. Surely, they will not then buy a football or something random. The same could be said in a situation where a chef has ingredients laid out on a countertop for meal preparation, such as lobster and pasta. We can assume the recipe may also call for butter, sauces, or seasoning to complete the recipe. We would expect that they will not add something random, such as window cleaner, to the countertop. In the same way, we can analyze predictive factors among social service workers to realize a desirable recipe for systems change based from the ingredients from Kania et al. [13], which include Policies, Practice, Resource Flows, Relationships and connections, Power dynamics, Mental Models.

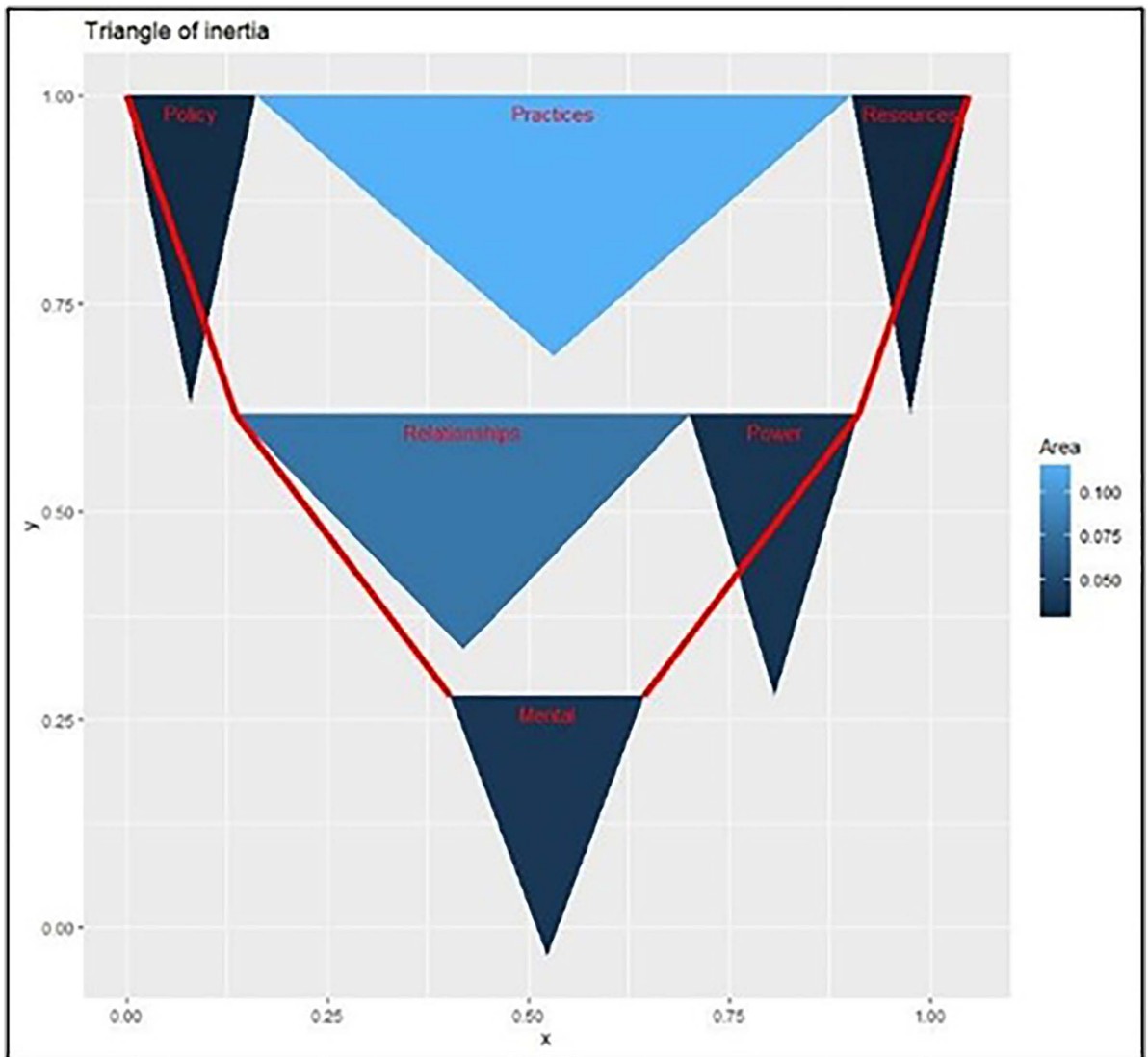

**Fig 2. Legitimacy of theoretical model for "Systems Change" from Kania et al. (2018) by theme according to data from RE.** ! RE!NSTITUTE™ and mathematical concepts from Bhaduri (2023). We interpret this as a realized version on the theoretical triangle shown in Fig 1. The shapes and the sizes of the components have been calculated using data from the 100-day challenges. Unlike Fig 1, the bases and the altitudes of the shapes here do convey statistical information. The bases represent how common changes in a certain category are (the wider the base, the more common the change) while the altitudes represent the inverse of the length of the confidence interval for this statistic (the taller the altitude, the more sure we are in the value of the base). This observed empirical triangle (Fig 2), therefore, confirms, in the process, the general shape of the theoretical triangle (Fig 1) in that the bases for "practices", "policy", and "resources", collectively (that is, "structural") cover the widest ground and the bases subsequently taper down on relational and mental changes.

## Dataset and procedural observations

Our dataset was sampled from RE!NSTITUTE™'s 100-Day Challenges™, which includes representative data sampled across the United States from states as diverse as California, Nevada, and Connecticut. Each agency in the respective locations reported what they experienced in a monthly report, which we explain in the text that follows.

To establish concrete manifestations of previous theoretical contributions on systems change, we first identify the frequency and the hierarchical order legitimacy for each of the following items: Policies, Practice, Resource Flows, Relationships and connections, Power dynamics, Mental Models. Doing so will allow for making predictive claims for each item regarding their interconnectedness among a web of scenarios. Data that RE!NSTITUTE™ collected is illustrated in Table 2 and the quantitative summary measures that result from them are shown in Table 1, according to the mathematical concepts from Bhaduri (2023). The most basic of all the metrics, support tells us the number, or equivalently, the proportion of times (if we want to normalize the number through a grand count) an "item" (in our shopping analogy) was purchased either on its own or in conjunction with others.

For us, the innovation types in the model of Fig 1 function as items. Take the item "policy", for instance. In the real data collected by the RE!NSTITUTE™, each instance (that is each row of Table 2), which shows the result of one social change experiment may be thought of as a "receipt" from a customer coming out of a supermarket after they have purchased items. The intersections on that specific row indicate the items that specific customer bought. Items such as "Policy" – in this sense – was "bought" 10 times either on its own or in association with some other item. This situation is shown in red font in Table 2. We have 62 of such "receipts". The support, therefore, for "policy" is 10/ 62 = 0.1613 (please see Table 1). This represents how high they correspond to "policy" on the bar plot in Fig 3. So, an item which is very popular (or, in our case, an innovation which is quite easy to bring about) would have high support. One which is unpopular or an innovation that is less common would have lower support.

To impart an exact (or data-based empirical) shape to the theoretical triangle (Fig 1) we began with support. In this way: innovation sub-types will be shown through smaller triangles within the (possibly) larger triangle – the bases of these sub-triangles in keeping with their respective supportive items in Fig 2. An innovation that is easy to bring about would occupy a large portion of the foundation: it would sit more solidly – enjoying a larger "share" at its appropriate level (or row). *Support* has its drawbacks. It does not, for instance, show how many times an item was "pure", that is, bought on its own (the following measures will fix such problems). This strategy has a few advantages, still. The most immediate one is bringing out the subdivisional asymmetries that the theoretical triangle could not. This means that if one looks at the conceptual structure (Fig 1), one gets the impression that each substructure within each level, covering an equal foundation and area, is as crucial as any other. That this need not be – and is not – the case is revealed by the actualized triangle (Fig 2), actualized through the collected data, actualized through the calculation of support, and agreeing on this triangle-base convention. Notice within the "structural" and "relational" levels, the ingredients occupy *different* amounts of foundational spaces (these spaces serving as proxies for how easy it is to bring about changes in that area).

## Descriptive analysis

In each of the subsequent sections, we will explain our general findings and then detail them. Firstly, we find that the theoretical interventions in Fig 1 by Kania et al. [13] are solid in their application. Secondly, we find that "practices" and "relationships and connections" are the most utilized interventions by the social workers in the 100-Day Challenge™. Thirdly, changes in "mental" structures are one of the hardest to bring about, evidenced through its small "support" value.

**Table 1. Support and confidence interval lengths summarized across features.**

| Feature | Support | Inverse of 95% C.I. length |
|---|---|---|
| Policy | 0.1613 | 5.5463 |
| Practices | 0.7419 | 4.6794 |
| Resources | 0.1452 | 5.7504 |
| Relationships | 0.5645 | 4.2265 |
| Power | 0.2097 | 5.0760 |
| Mental | 0.2419 | 4.6795 |

**Table 2. Data collected by the RE!NSTITUTE™. Each row represents one deployment of the 100-Day Challenge™. A cross indicates changes in the corresponding aspect could be brought about in that instance of the experiment.**

| Innovation | Policy | Resource | Practice | Relationship | Power | Mental |
|---|---|---|---|---|---|---|
| Structural | | | | X | X | |
| Structural | | X | X | | | |
| Structural | | | X | | | |
| Ways of Working | X | | | X | | X |
| Technological | | | X | X | | |
| Structural | X | | X | | | |
| Ways of Working | | | | X | | |
| Technological | | | X | | | |
| Ways of Working | | | | X | | |
| | | X | X | X | | X |
| Ways of Working | | | X | X | | |
| Structural | | | X | X | X | |
| Structural | X | | | X | | |
| Ways of Working | | | X | X | X | |
| Structural | | | X | X | X | X |
| Structural | | | X | X | X | X |
| Structural | | | X | X | | |
| Structural | | | X | X | | |
| Technological | | | X | | | |
| Structural | | | X | | | |
| Structural | | X | | | | |
| Technological | | | X | | | X |
| Ways of Working | | | x | | | |
| | | | X | | | X |
| Technological | | | X | | | |
| Technological | | | X | | | |
| Structural | | X | | | | |
| Structural | | | X | X | | |
| Ways of Working | | X | X | X | | |
| Structural | X | | X | | | X |
| Structural | X | | X | | | x |
| Structural | | | X | X | X | X |
| Structural | | X | | | | |
| Structural | | | X | | | |
| Technological | | X | X | | | |
| Technological | | | X | X | | |
| Structural | | | X | X | | |
| Structural | | | | X | X | |
| Ways of Working | | | X | X | X | |
| Technological | | | X | X | X | |
| Ways of Working | | | X | X | | X |
| Structural | X | | X | X | | |
| Structural | | | X | X | | |
| Technological | | | X | X | | |
| Ways of Working | | | X | X | | |

*(Continued)*

**Table 2.** (Continued)

| Innovation | Policy | Resource | Practice | Relationship | Power | Mental |
|---|---|---|---|---|---|---|
| Structural | | | X | | | |
| Structural | | | | X | X | X |
| Ways of Working | | | | X | | X |
| Structural | | | X | X | X | X |
| Structural | | X | X | | X | |
| Structural | X | | X | | | |
| Structural | X | | X | X | | |
| Technological | | | X | | | |
| Technological | | x | X | | | x |
| Structural | | | X | X | | |
| Structural | X | X | | | X | |
| Ways of Working | | | | X | | |
| Structural | | | X | | | X |
| Structural | | | X | X | | |
| Structural | X | | | | | |
| Structural | | | X | | | |
| Ways of Working | | | | X | | |

Table 1 above shows the point estimates and the corresponding heights for all these sub-triangles. The area of any triangle, being a combination of its base and height (0.5xbasexheight), may be big if, either innovations in that regard are evidenced to be more common (that is, it has a large base, i.e., "support"), or, if we are very sure of our guess for this "support", whether big or small (that is, if the interval is compact). For instance, in our previous furniture-house example, with supports, i.e., bases of 0.9 and 0.01, if we are equally sure of these values (say, 1), the area of the "furniture" component will be 0.45 (=0.5x0.9) and that of "house", 0.005.; furniture, thereby exerting a greater influence on how the "furniture-house" triangle looks. In this sense, for our real 100-day challenges, "practices" has the biggest share among all the sub-triangles.

## The model stands

Our first major finding is that Fig 2 supports the theoretical beliefs in Fig 1. We demonstrate this through adding the support fractions. Observe the grand support for the "structural" level with contributions coming from "policy", "practices" and "resource flows" is $0.1613 + 0.7419 + 0.1452 = 1.0484$. While the grand support for the "relational" level with contributions coming from "connections" and "power dynamics" is $0.5645 + 0.2097 = 0.7742$. That for the "mental" level, with only one component, is 0.2419. Now

$$1.0484 \; > \; 0.7742 \; > \; 0.2419.$$

Were those numbers some other way, that is, if this *order* were disturbed, the *stability* of the composite triangle would have been compromised. The explicit (structural) level may not have (inversely, if we agree to the inverted arrangement) supported the semi-explicit (relational) level, which, in turn, may not have supported the implicit (mental) level. A neat hierarchy would have been lost and the levels may have toppled over, destroying the prevailing stable equilibrium. We observe how, within each broad level (especially "structural" and "relational"), the components ("policy", "resources", etc.) seem to compete for a bigger share of the base. It is much akin to these broad levels having a (hierarchical) determined sequence of shares and the contributing ingredients vying for maximal control within that level, the less common innovations having to compromise and compensate (through having smaller shares), to respect the pre-determined shares of the

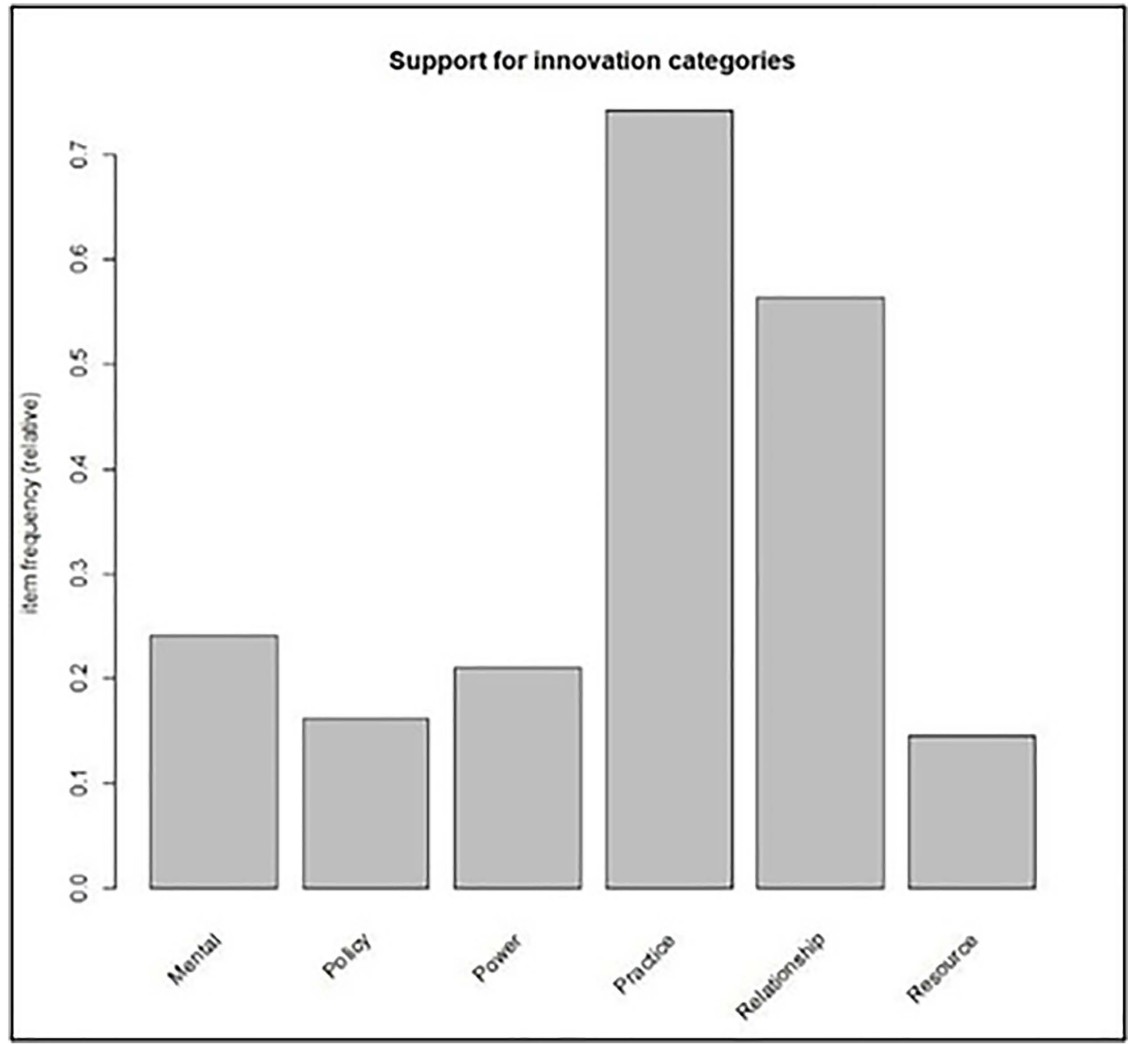

**Fig 3. Frequency of change by theme according to data from RE.** !NSTITUTE™, based on a theoretical model for "Systems Change" from Kania et al. (2018) using mathematical concepts from Bhaduri (2023). The heights of the towers describe the rate at which changes in the corresponding types were witnessed to occur. For instance, of all the 100-day challenges studied, we found about 55% of times changes in "relationship" were observed, either changes in "relationship" only, or in "relationship" along with those in other types.

larger categories. We bring out how changes in certain interventions are substantially and consistently more common than others.

We notice next the theoretical model (Fig 1) doesn't fully exploit the power of the pieces' heights: the sectors contributing to each broader level are equally tall. The heights are not quite taken to *represent* or *signify* anything concrete. This, to us, is a waste of precious cosmetic promise. An inferential possibility gone abegging. One might transmit crucial information through *differing* heights. This is where the collected data, again, helps us. And the information we choose to transmit is the length of the confidence interval for the corresponding "support" pieces. This length is a proxy for the reliability of a "support" estimate.

This is vital for many reasons, mainly because of the need to store a faithful record of the *sampling fluctuations.* This stems from an acknowledgement that the data we collected forms just a part – that is, a sample – of a larger population

that is "out there", beyond our reach. This means as our data change, i.e., as similar 100-day challenges get designed, the support or the confidence numbers could potentially change with them. And we need to ensure the fluctuations are not extreme enough to render the current results pointless. Concretely, those similar (possibly neighboring) communities that we didn't implement (but on which we could have implemented, in theory) the 100-Day Challenge™. This *statistic* we observed, say the specific 0.1613 "support" value for "policy", is just one out of infinitely many such values that could have been possible, were the RE!NSTITUTE™ to collect data from similar, but other such communities. Stability or reliability, therefore, becomes a natural worry. Had 0.1613 changed to a number drastically different, say, 0.9217, under a different sample (that is, had the *sampling fluctuation* been substantial), our confidence in saying that 0.1613 is a good guess for the true unknown "support" (in the population) would have gone down. A confidence interval gives us a range of values in which the true unknown "support" (in the population) may lie. 95% confidence intervals are made, for instance, in a way so that if someone were to replicate the 100-Day Challenge™ 100 times in the potentially different communities and calculate these intervals, around 95 of them would cover the true unknown "support". The shorter these intervals, the more reliable our guesses – like 0.1613 – are. Technical details on the calculation of these intervals will be distracting to our main theme.

We point readers interested in the interval calculations to [30]. Through these technicalities, however, the 95% confidence interval for "policy"'s "support" turned out to be [0.092931414, 0.27323788], generating a length of 0.2732–0.0929 = 0.1803. The height of the "policy" triangle in Fig 2 is the inverse 1/0.1803 = 5.5463 of this length, designed to reflect how frank we are in the point estimate of 0.1613 (please see Table 1). The bigger this height, that is, the shorter the length, the more sure we are in this guess. In contrast, the 95% confidence interval for "practices" is [0.606897053, 0.82057017], giving a height of the triangle as 1/(0.8206–0.6069) = 4.6794. We are therefore a little surer in our "support" guess of 0.1613 for "policy" than our support guess of 0.7419 for "practices".

We have, therefore, achieved what this subsection primarily set out to do. Offering a concrete shape to the conceptual triangle. We took the shape to be a composition of two aspects: the bases of the triangles (these are taken to be "support"s, showing how prevalent the innovations are), and their corresponding heights. The inverses of the lengths of the confidence intervals are taken as descriptions of the support values' reliability. Putting these two properties together, we have erected data-driven triangles, improving the theoretical triangles, in the process.

## Dashboard: Predictive possibilities under incomplete knowledge

Through "support" – and metrics related to "support" such as its confidence interval – one can, as shown in the previous section, estimate an exact shape of the conceptual triangle. What concerns us now is beyond a specific shape: it is checking whether one can, based on *observed* changes in certain habits or ways of work, *predict* whether changes in other traits are likely to be triggered. This is not unlike how recommendation systems work [31]. For a shopper with some candles and a birthday cake in his shopping trolley, we are reasonably sure he's going next to buy a birthday card. We can even recommend such a card instead of something that doesn't "go with" the two observed items (i.e., some odd item such as chicken soup). For example, in our context, the {candle, birthday cake -> birthday cards} automacy will take on a similar form, once we treat items as habits.

Concretely, we reinforce Lipsky's [6] theories on street-level bureaucracy, and are curious to know whether automatic and reliable pathways like {relationship, practice, power -> mental} exists (this specific one – as shown below – does) through which, one may claim, if through certain innovations like removing bureaucracy and "red tape" [4], changes in relationship, power and practice are guaranteed, changes in mental models are extremely likely to be triggered. We need our next technical idea, that of "confidence" to quantify this quandary.

## Confidence

With the seen items like {candles, birthday cakes} or {relationship, practice, power} taken as "antecedents" or "if" items, and the unseen ones like {birthday cards} or {mental} as "consequent" or "then" items, *"confidence"* quantifies how often

the "then" items are seen in conjunction with the "if" items, relative to how often those "if" items are seen, either on their own or in association with others. This fraction, then, estimates how automatically the "if" set will lead to the "then" set. As an example, to work out the confidence behind the path (sometimes called a rule) {relationship, practice, power -> mental}, we see that the "if" items occurred eight times, shown in red, in Table 2. That is, on eight 100-Day Challenges™, due to the innovations, changes in at least relationship, power and practice were brought about. Out of these eight, there were four cases where changes in mental models also occurred. The confidence behind the {relationship, practice, power -> mental} path is therefore 4/8 or 0.5. This is how often mental changes correlate with those in relationship, practice and power.

Instead of recording a tedious list of many such possible relations and their associated confidences, we offer our readers an interactive dashboard at https://moinak.shinyapps.io/MarketBasketDashboard/ where they can set tolerances for support and confidence to explore reliable connections.

These connections – we stress again – represent correlational links for the moment and need not be taken as causal pathways. In our previous works – [31–34] – we have explored this matter of causality in analogous social settings using similar network structures – Markov random fields – to showcase how gender of respondents and political leanings causally sway their opinion towards current business practices, how students from diverse backgrounds express their sentiments towards the living conditions of migrants at the US-Mexico border, etc. The estimation of these random fields, however, depends on the delicate balance of discerning false positives and false negative edges through regression techniques such as lasso which are data hungry algorithms. In view of the fact that this is the first of a long sequence of ongoing 100-day challenges, data, though of high quality, are relatively scarce, and we have refrained from querying causal effects. In time, as similar high-quality data become available, we can deploy the Markov random fields much along the lines of the works cited earlier to explore more interesting causal pathways.

The networks shown on the various subpages are intuitive. The figure (Fig 4) above, for instance, shows various ways – some reliable (that is, having a high "confidence"), some not so reliable (with a low "confidence") of possibly bringing about changes in "mental" structures. Since "mental" always, in this case, is the result – the product – of the prediction, the arrows leading to "mental" are shown in red. The predictive ingredients ("resources", "policy", etc.) are shown in blue rectangles while the "rules", that is the way of predicting, are shown in various shades of red. The deeper the red, the more sure we are of the fact that changes in "mental" structures will be brought about. If one hovers on "rule 12", for example on the dashboard, which is deeply red, a "confidence" of 0.5 will be shown. One unpacks it this way: if a study is designed that guarantees changes in "relationship", "practice", and "power" (note these pieces go on to make the predictive ingredients of rule 12), changes in "mental" structures (the prediction from rule 12) will be promised with 50% chance. This "confidence" drops to 0.111, for instance, with rule 1 (shown through a fainter red): a simple – and relatively more unreliable promise – that changes merely in "resource" will correlate with changes in "mental" structures.

Among the various recipes or strategies for achieving transformative change, the most effective ones initiate change through the modification of practices and relationships. This approach is rooted in the concept that significant changes in mental models, or the underlying mindset and buy-in from workers, are best achieved by first altering the day-to-day practices and the nature of interactions between individuals. When processes and social dynamics are transformed, they create an environment that facilitates the natural evolution of mental models, leading to genuine and sustainable change in the organization.

Shifts in relationships and practices are foundational to transformative change. By focusing on modifying how workers interact and carry out their duties, these strategies ensure that the groundwork is laid for deeper cognitive and psychological shifts. As employees experience and adapt to new ways of working and engaging with their peers, their mental models begin to align with the new norms and expectations, making transformative change not only possible but inevitable.

Networks – the crucial ingredients in market basket analysis algorithms – are notoriously difficult to estimate and manipulate due to their fundamental non-Euclidean-ness, in case the sample size becomes unmanageably large. While

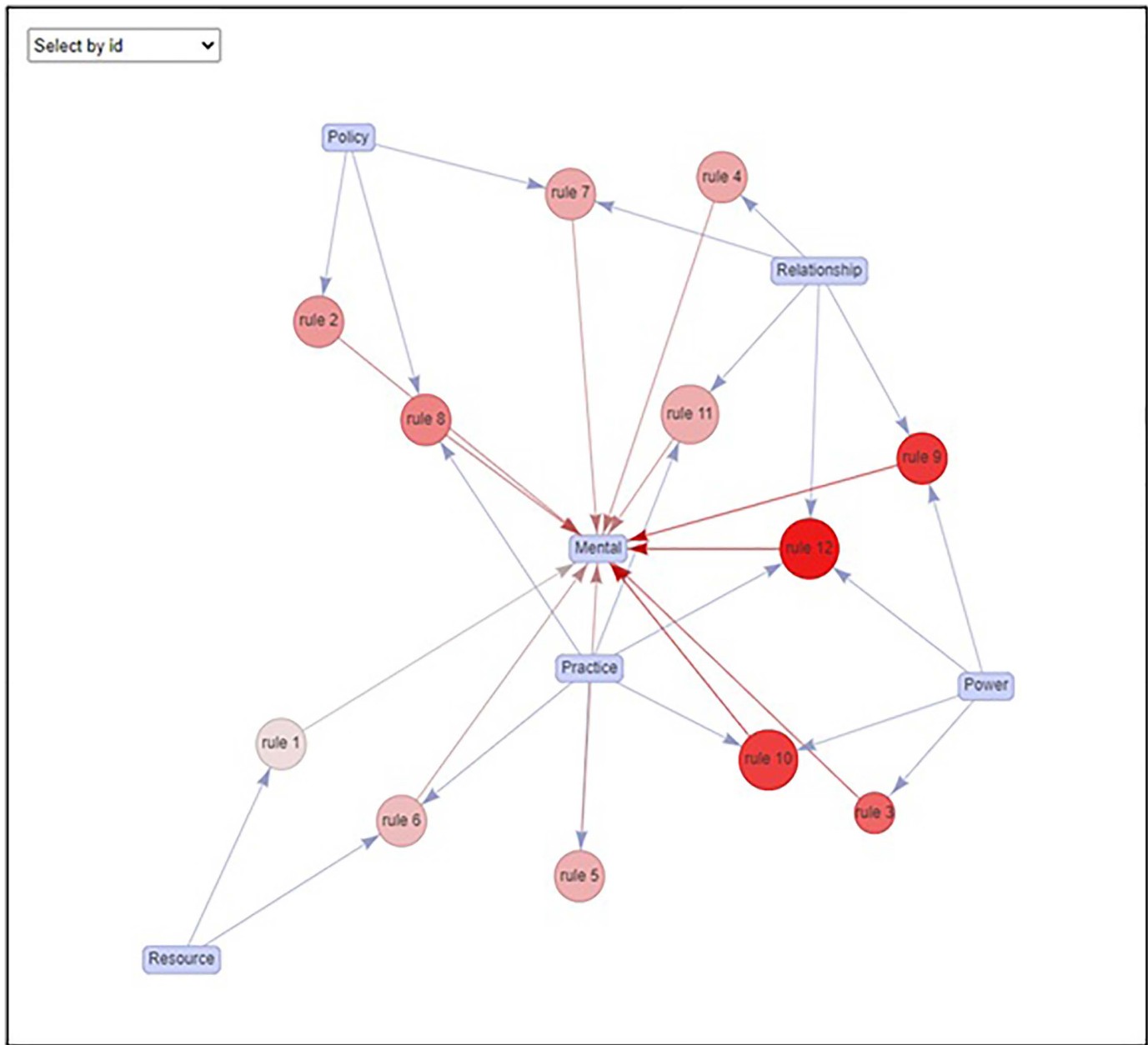

**Fig 4. A typical map of likely connections that lead to changes in mental structures.** The more likely connections are shown in deeper red. We invite readers to explore other possibilities (for instance, those connections that lead to non-mental targets) on our interactive dashboard. One implements these rules in an input-output sense. If changes in the input details – shown by the blue arrows – are promised, one is given the rate at which changes in the output detail – shown by the red arrow – accompany. The higher this rate, that is, the more (correlationally) automatic the input-output connections are, the darker the shade of the red node representing the rule. For instance, rule 12 says if one designs a relocation study that generates changes in "relation-ship", "practice", "power", changes in "mental" structures will be promised with a reasonably high chance (50% if one hovers on the dashboard).

for the current study the sample size is not a concern, we anticipate future waves of 100-day challenge data which could make the algorithm slow or less interpretable. To combat the size problem, we can draw sequential subsamples (while maintaining global network invariants such as the asymmetry of the degree distribution) and create bootstrap intervals to

quantify variations in statistics of interest such as the support or the confidence. The approach is becoming standard in the literature. We point interested readers to [35] where similar algorithms were implemented to fatal crash data in large jurisdictions."

Once relationships and practices are successfully altered, the likelihood of enduring transformative change in mental models increases significantly. The changes in these fundamental aspects of the work environment serve as catalysts for employees to reassess and eventually embrace new mindsets. Over time, as the consistency and effectiveness of the new practices and relationships become evident, mental buy-in from the workers becomes stronger, leading to a comprehensive and lasting transformation within the organization.

We note another fact worth mentioning. The predictions done through "confidence", that is, the laying out of a likely "then" item, are possible – as the {relationship, practice, power -> mental} rule shows – even under *incomplete knowledge of the predictors*. What we mean by this is, we need not have total knowledge of all the other *potential* predictors to predict the chances of the one item left out. For instance, in the {relationship, practice, power -> mental} rule above, we need not know states of the other potential predictors, that is, whether changes *were* or *were not* brought about in "resources" and "policy". This effectiveness under partial knowledge remains a key aspect that makes these association rules especially attractive.

The chief intent behind the current study was to explore correlational connections between and among working components, and shed quantitative light on the strength of those connections. It was not to decide whether policy changes would better one specific individual's chances with or without health issues, for example (causal connections here would necessitate the bringing in of A-B tests and uplifts). Covariates such as health or substance abuse could be potentially vital predictors to guess homelessness (or recurring homelessness). In future runs of the 100-day challenge, we plan to incorporate these details to offer both population-level and instance-level recommendations.

Network objects are underexplored in the social sciences. Consequently, the dashboard, which brings out specific kinds of network structures, while extremely insightful, must be interpreted with some caution, in case one is unfamiliar with non-Euclidean mathematics (we direct such readers to [36], for instance). For instance, contrary to what one typically encounters in a graph, in these structures, there are no horizontal and vertical axes with set roles. As a result, each time one visits the dashboard, there is a slight possibility that this structure may be rotated some different way (although every attempt has been made to prevent this from happening through the Force Atlas algorithm [37]. What matters more, however – the strength of the connections, that is, the rules shown by degrees of red – will stay unchanged. In case a rotated representation results, it is vital, therefore, to acknowledge that the correlation-based recommendations – statements that show changes in which property accompany changes in which others – have not altered. There are other metrics such as the lift ratio through which one may choose to represent the strength of the rules. In the present version of the dashboard, we have refrained from these reports since they will be distracting to the main theme of the current study. With our currently ongoing follow-up studies (under "Future Work"), we are going to supply a variety of such metrics along with suggestions on which might be apt under which circumstance.

## Spotting changes in continuity and measuring their impacts

Establishing the priority of ingredients in a systems change allows us to move next to explaining performance. Since RE!NSTITUTE™ records the success outcomes from all associated agencies and their geographic location during their 100-Day Challenge™, we can consider how success changes over time and compare between agencies and location. This allows us to identify potential barriers and catalysts for success during the 100-Day Challenges™ for a systems change to occur. The methodology we demonstrate are implemented, for now, on sixteen geographical locations that the RE!NSTITUTE™ conducted their study. But we stress that it may be extended to similar studies over other regions to launch a larger comparison in the future.

In the following section, we will show how success changes over time in Northern Illinois and Desert City. We then compare among counties across the United States to assess similarities and differences by utilizing comparative analyses techniques, which spotlight counties with like tendencies and ones that diverge in practice.

## Success measures

Here we query the impact *time* plays on the 100-Day Challenge™ evolution, allowing us to consider how the rate at which people were housed by social service workers changed (i.e., did they pick up or slow down). We compare reported data to the potential numbers to consider what *could have been achieved* had there been no shift to the actual numbers that were *observed*. We suggest that the difference could be attributed to a lowering of the force or the impetus that drives the way the relocation rate series is channeled forward leading to a lull or a surge in housing people who experience homelessness.

Such mathematical strategies have been demonstrated to work in a variety of settings. The gap between what *is* and what *would have been* exploited to measure the impact of COVID waves [38], the interestingness of cricket and soccer matches [39] or other types of sports such as mountaineering [40]. Aside from quantifying this way the estimated impact of the potential causes, one may, through these "change-points" check the effectiveness of the grand 100-Day Challenge™ structure as a whole.

The responses could be immensely impactful. One could, for instance, reliably unearth the reasons behind these shifts during the challenge (i.e., workers having to miss work). While our two previous subsections (legitimizing the hierarchy and discovering strong connections among innovation types) considered the 100-Day Challenge™ in total, now we attempt to be more granular as the challenge proceeds, either in retrospect or online. We ask, for instance, not eventually, but *as the challenge goes on*, whether there were phases where the rate at which the housing of people experiencing homelessness picked up or slowed down. The relevance of the 100-day deadline – and the relevance seems palpable (notice how the change points indicating upped rates line up towards the end of the experiment on Fig 5, for Northern Illinois, maybe checked when we compare the time series of relocation rates from this deadline-based (imparting greater urgency) exercise to those where no deadline was imposed.

The technicalities needed originate from theory of point processes [41] or, precisely, *Poisson* processes [42] and change detection algorithms introduced by Bhaduri [43].

We point readers interested in mathematical technicalities to a crucial appendix (Appendix C in S1 File) we have designed for this purpose using multiple testing procedures [31,43] and tessellations [44]. In summary, we track the times at which homeless people get relocated, construct a number (detailed in appendix C in S1 File) that serves as the rate of relocations, and used this number to track the earliest instance of a changed rate over the course of the 100 days. Once this earliest instant is found, we begin the algorithm afresh to detect the next possible point(s) of change. Patterns in these detected "change" times hold particular relevance, especially in tracking how time-bound relocation exercises such as the 100-day challenges progress, with and possibly ascribe the gap without using external (such as governmental) supervision. For instance, with time-bound exercises, a stronger concentration of such change times may be present towards the end of the fixed period which could reflect workers' upped urgencies. One may compare the efficacy of two time-bound attempts with how similar their change points were, or compare one time-bound study to a general relocation attempt. These "change" times also enable us to measure the impact of a shift. We can contrast the forecasts from the pre-change piece (had there been no change) to the observed relocation numbers from the actual, post-change piece. And possibly ascribe the gap – using field of knowledge – to preventable causes such as absenteeism. We point readers interested in further technicalities to [45–54]. The strategy described above helps pinpoint where changes occurred, that is, deliver solid definitions of the pre- and post-change pieces, crucial for these gap measurements. We look at two specific examples below to elaborate these matters.

Below, Fig 5 describes the result of implementing this algorithm on Northern Illinois and Desert City. These vertical separators represent "change-points", i.e., times around which the rates at which people were relocated, get altered

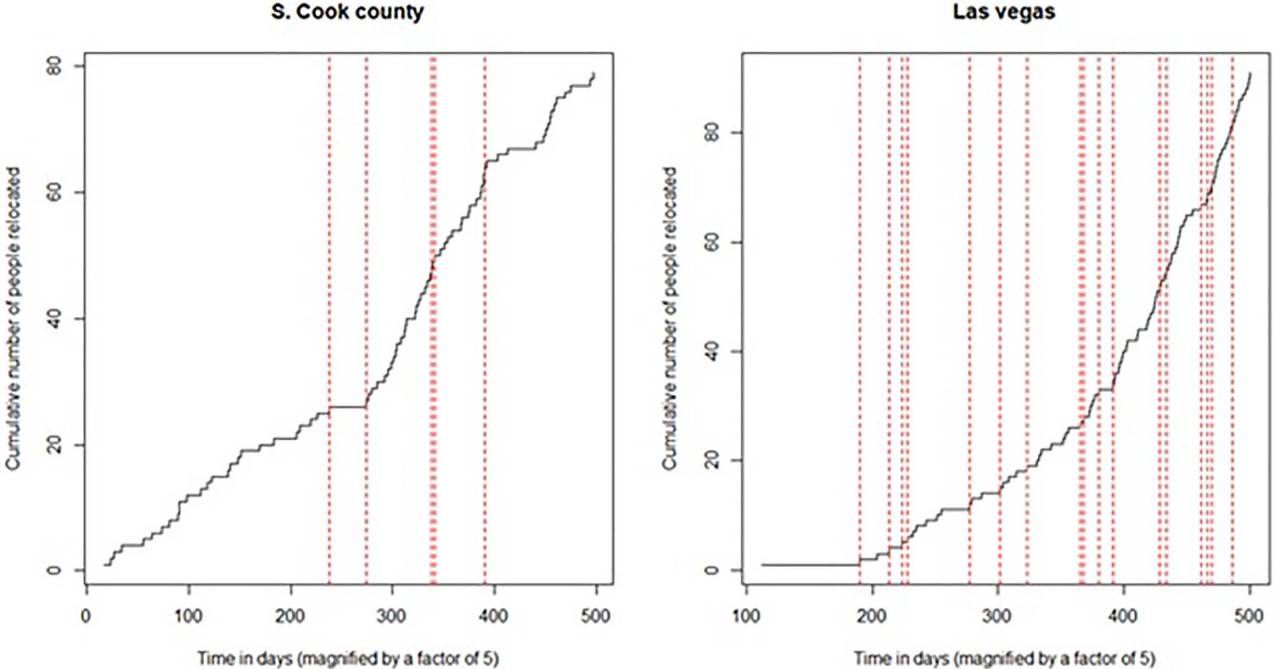

**Fig 5. Epochs of stable relocation rates are shown on either side of the red vertical separators for (a) Northern Illinois and (b) Desert City.** Each vertical separator represents an estimated change-point, that is, a point in time over the course of the 100 days during which the rate of relocating homeless people improved or deteriorated substantially. For instance, had the change-point around day 80 (=400/5) not been there for Northern Illinois, the total number of people relocated could have been 85 instead of 80. Crowding of change points towards the end of the experiment is observed for nearly all regions.

substantially. The time axis (for technical reasons demonstrated in, for instance, Bhaduri [35] is stretched by a factor of 5. For instance, the last break-point in Northern Illinois —around the 400-point mark corresponds roughly to the 79th day (on Table 3, the highlighted red).

Results of a larger scale study across many such places are condensed in Table 3. We find for most cases, the majority of change-points are towards the end of the 100-day period. This could possibly be a reaction to increased urgencies. However, here we content ourselves with reporting this fact and leave the explanations to those more qualified to comment.

Once we have these "change" answers, i.e., lines defining stable phases of constant relocation intensity, the next thing we can do is measure the extent of change, as one moves across a boundary. Let's take the penultimate stable phase for Northern Illinois, for instance, the one lasting over 350–400, roughly. The gradient was quite steep here, i.e., the rate at which people got relocated was quite high. For whatever reason, this progress was hampered post-400: the rate dropped substantially. The measuring of the change impact can be done this way (details in Bhaduri [38]: we would let the [350–400] phase carry on a bit longer (through a process called point process bootstrapping) to represent what would have happened had there been no change. Maybe in that case, the vertical axis would have stretched to 90. So, this gap between what is (80) and what would have been (90), summarizes the extent of the jolt: We missed the opportunity to help 10 individuals receive housing. Explanations may vary, especially when we consider similar instances beyond Northern Illinois County.

So far, we have established that the strategic implementation for systems change involves relationship building across departments and agencies to establish new practices and procedures to streamline flows of success, which

**Table 3. Records of days (within the 100-Day Challenge™ period) around which substantial changes occurred in the rate at which people were relocated.**

| Place | Detected change-points (in days) |
|---|---|
| Northern Great Lakes | 39.8, 67.2 |
| Northern Illinois | 47.6, 54.8, 67.6, 68.2, **78.2** |
| Desert City | 38.0, 42.6, 44.8, 45.6, 55.4, 60.2, 64.6, 73.0, 73.6, 76.0, 78.2, 85.6, 86.6, 92.2, 93.2, 94.0, 97.2 |
| Northern California | 5.0, 8.4, 10.2, 13.8, 15.4, 15.6, 25.0, 26.2, 26.6, 27.0, 29.2, 35.4, 36.8, 55.8, 56.8, 61.8, 62.8, 63.6, 64.2,65.2, 65.8, 66.2, 66.8, 68.2, 68.8, 69.4, 70.4, 71.4, 71.8, 72.6, 74.4, 74.8, 76.6, 78.8, 82.2, 83.2, 84.4, 85.8, 86.6, 88.6, 89.4, 92.2, 97.0 |
| Southeastern Florida | 37.4, 38.2, 38.8, 39.4, 39.6, 39.8, 40.0, 40.2, 42.2, 44.2, 44.4, 47.4, 47.6, 49.0, 49.8, 50.8, 51.0, 61.8, 64.2, 66.6, 77.6, 85.4 |
| Western Connecticut | 63.8, 78.4, 79.4, 82.0, 84.6, 86.6, 92.2, 97.2 |
| Coastal Connecticut | 37.4, 41.2, 54.6, 67.4, 78.6, 90.6 |
| Northeastern New England | 42.4, 44.8 |
| Central Connecticut | 51.0, 52.8, 60.0, 62.2 |
| Southern Connecticut | 37.2, 70.0, 71.6, 73.2 |
| Central Connecticut Capital Region | 48.0, 49.8, 50.4, 51.0, 51.6, 52.8, 53.0, 54.4, 57.6, 58.2, 60.8, 62.2, 63.0, 63.4, 63.6, 82.6, 83.2, 87.0 |
| Southwestern Connecticut | 76.2, 78.4, 80.4, 80.6, 82.0, 82.6, 96.2 |
| Central New England | 27.8, 64.6 |
| Silicon Valley | 27.2, 44.4, 45.8, 46.4, 53.4, 55.8, 56.6, 68.2, 72.4, 80.6, 81.4, 85.4, 87.0 |
| Pacific Northwest | 27.6, 34.2, 36.2, 37.2, 37.6, 37.8, 39.8, 40.2, 41.4, 43.8, 45.4, 46.4, 47.4, 48.2, 51.8, 55.2, 60.8, 62.2, 82.6, 84.0 |
| Northeastern Illinois | 63.2, 66.2, 75.4, 87.8, 89.0, 90.8, 95.0, 96.8, 97.8 |

could explain surges in success rates. The drops that follow are the result of when the stream is disrupted. Moving forward, participating agencies may build their relations and implement procedural changes in varying ways that can be unique or resemble other projects. We now turn to comparing a range of 100-Day Challenge™ projects across geographic locations.

## Comparing relationships across geographic locations

Knowing the prioritization of essential parts of systems change, along with considering changes in the rates of success over time, we transition now to establishing similar patterns of behavior between participating agencies in the 100-Day Challenge™. We broaden our scope to consider projects across geographic location in the United States to understand the likeness among the various strategies that were implemented by workers in each location.

Bhaduri [38] shows how change points can be used in the context of the COVID outbreak: what constitutes a "wave", how to measure its enormity, how to group countries through them, etc. One may follow, more or less, a similar approach. Similar change-points would make two regions become connected on a network such as the one in Fig 6. A wealth of predictive machinery from statistical network science may be deployed subsequently. There is an edge (i.e., the change-points from the studies on these two regions occurred at similar times), for instance, between Northern Illinois and Desert City but none between Connecticut Central and Midwest City.

We point out in the previous runs of the 100-Day Challenge™, the potential reasons behind the possible shifts were not recorded for all the counties. The experiences, such as the ones shown above for Desert City and Northern Illinois, were anecdotal and based on recollection. This impedes us from carrying out a fuller analysis in this section on change-points. In future runs, however, a thorough record will be kept. In the meantime, we intend for the steps outlined in this section to be meaningful to those intending to design similar studies, whether among the social service and homelessness context or beyond.

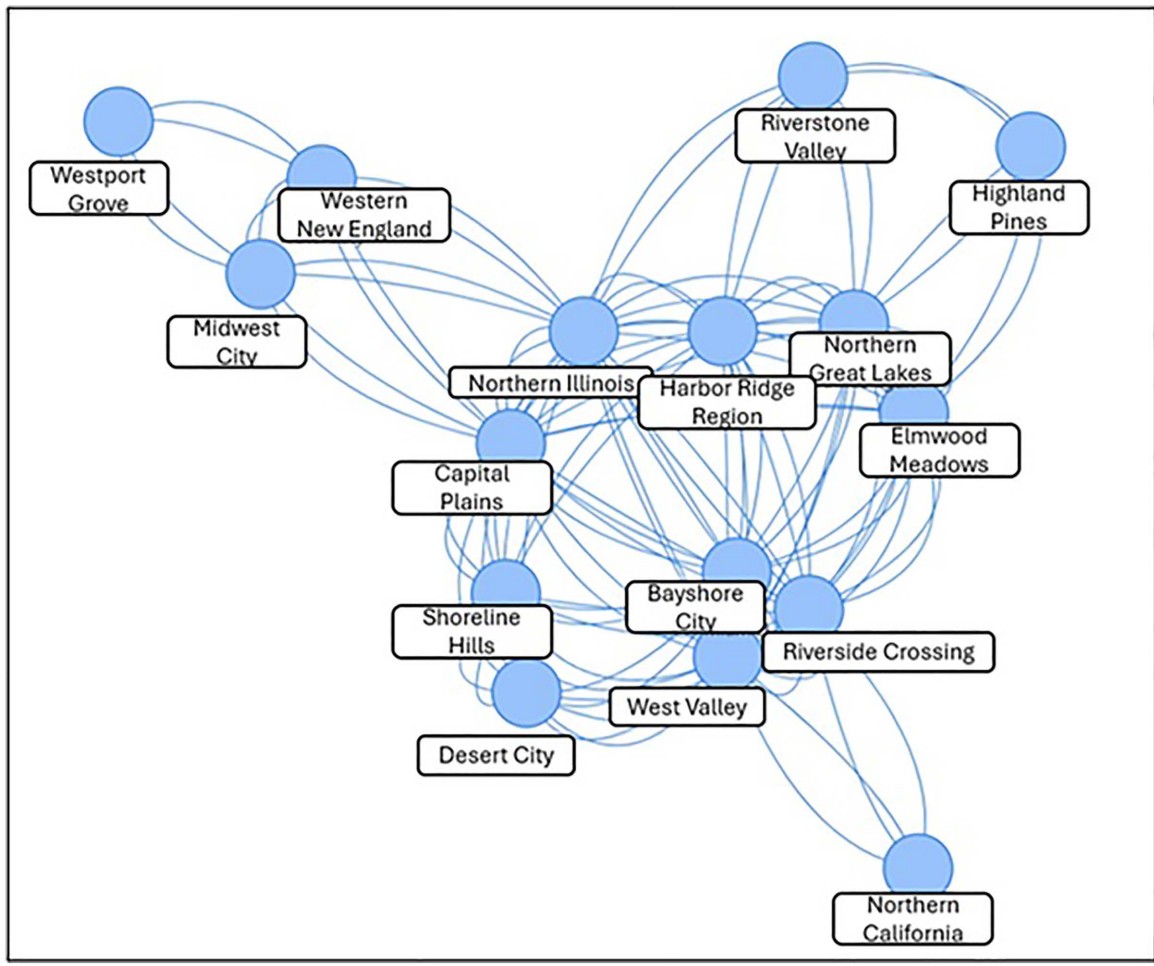

**Fig 6. Network joining regions with similar change points.** If two cities experience drastic shift in relocation numbers around similar times over the 100 days, there will be an edge joining them. Else, there will not be one.

## Summarized Findings

This study has evaluated components for systems change, based from theoretical summations from Kania et al. [13], by utilizing data from RE!NSTITUTE™'s 100-Day Challenge™. Firstly, our descriptive analysis shows the legitimacy of the hierarchy in Fig 2, that Policy, Practices, and Resources bring about or at least correlate with changes in relationships and power, thus leading to mental changes. Had the arrangement been configured in any other way, there would be no stability in the model and it would fall apart, essentially. We find that the integrity of the theoretical model is significantly sound.

Secondly, we find that among the theoretical components in Kania et al.'s [13] model, (policies, practice, resource flows, relationships and connections, power dynamics, mental models), front-line workers enacted changes in practices (i.e., how they carried out the demands of their job) and changes in relationships and connections (i.e., broadening their network and increasing the quality of their network) most frequently to successfully make a systems change. While many typically focus on the lack of resources allocated to services, such as changes in funding, workers broadened their social networks across departments and agencies while improving the quality of their relationships with other workers, also changing the routinized practices of their daily workload to achieve success during the 100-Day Challenge™.

Thirdly, we offer a dashboard at https://moinak.shinyapps.io/MarketBasketDashboard/ which showcases a variety of ways in which transformative changes in any aspect is correlated with changes in the remaining areas. This is revealed through a vast web recording the various possibilities. There are various recipes, but we find the one most significantly achieving transformative change through shifts in mental models (i.e., mental buy in from the workers) is first through strategies that change practices and relationships. Once shifts in relationships and practices take place, transformative change is likely among mental models.

Fourthly, we quantify our claims through offering solid numerical estimates. For instance, we observe if changes in power dynamics, practices and relationships occur, changes in mental structures occur nearly 50% of the times. The nodes with deeper colors on our dashboard represent connections that are more automatic (i.e., links that are more prone to occur or are more likely) than those represented by fainter ones. We can, for instance, also say that if we have shifts in resources and practice, then we can guarantee a mental change with a confidence of 1 (100%), assuming we have adequate workers. Alternatively, we see that when *only* resources are increased (i.e., increased funding), then we do not see a systems change. It's not enough. We also see the same with changing policy, alone. We cannot expect a systems change based from only one of these ingredients. Instead, we show the full recipe for systems change, assuming the presence of sufficient workers. We want to emphasize that this does not mean predictive qualities that guarantee an outcome, rather a recipe for administrators and policymakers to consider. In other words, the success of our recipe still assumes the presence of competent workers. Conversely (and more agreeable to our own views), having *only* skilled workers without a clear recipe makes expectations for a desired outcome difficult. Thus, we see programs like the 100-Day Challenge™ as crucial given that workers may disagree or see limitations of existing policy, as Lipsky [6] says (Smith [4]). We believe our contribution is crucial to systems change, especially among social service work.

Fifthly, we show surges in success over time in Northern Illinois Country and Desert City during a 100-Day Challenge™, along with stunts in such growth. We suggest that the difference could be attributed to a substantial altering of the chief underlying thrusts that drive the relocation mechanisms (i.e., housing people who experience homelessness). For instance, as Fig 5 shows, had the rate of relocation prevalent in S. Northern Illinois County prior to 80 (=400/5) days continued (i.e., had the change-point representing a drastic shift been absent), the curve could have increased instead of stagnating, helping relocate nearly 15 more people. It is unclear why (i.e., it is difficult to precisely point out the main reasons behind the lull).

Lastly, we can observe likeness among various locations throughout the United States that have participated in the 100-Day Challenge™. We see that Northern California stands alone, while Westport Grove, Western New England, and Midwest City share similar change tendencies. Capital Plains and Highland Pines share some patterns while Northern Illinois, Elmwood Meadows, Desert City, Riverstone Valley, Shoreline Hills, Northern Great Lakes, West Valley, Bayshore City, and Riverside Crossing share the most in common.

## Discussion

Our findings confirm many perspectives on street-level bureaucracy and theories on systems change. When ground-level workers are given the opportunity to conduct the functions of their job independent of oversite from administrators, in this case by participating under the framework of RE!NSTITUTE™'s 100-Day Challenge, workers improve their service outcomes and their productivity increases.

We can confirm the validity of Kania et al.'s [13] collective modeling of theories on systems change. We have successfully predicted systemic change, which can assist in formulating strategies. We also believe that this and our other findings means we have contributed, objectively and quantitatively, to answering Coffman's [10] call-to-action by testing and confirming the validity of theories on systems change. We are providing recommendations that hold a promise of bringing about certain kinds of systems change if certain other kinds are guaranteed. This is achieved by pointing out apt conditional correlations.

In our analyses, the most frequent or common ingredients in the model that lead to a systems change are "changes in practices" and "relationships and connections" in leading to the goal, which is transformative change. Our finding that relationships and connections (networking) are the most influential ingredients reinforce the findings of the Nicholson Foundation [11] and Wallace and York [15], which we find to be valuable insights for policy workers. The increased successes experienced by cross-agency collaborations reduces service fragmentation. However, we offer exploitable nuances and subtleties through judicious application of mathematical techniques, such as Market Basket analysis. Fig 2 and Fig 3, found through these analyses, spotlight that all the items in each row of the model are not equal. We point out how even within a large and general organizational category such as the structural, changes in certain subcategories (such as practices) are more common, or potentially easier to bring about, than certain other subcategories (such as resources) within the same organizational categories. Fig 2 shows how the confidence between items varies with changes in policy, resources, and power being the most impactful.

While changes in these items carry the most significant weight in changing systems, they are not always the first strategies many think of. This is probably due to the lack of feasibility for workers to change their resources (funding) or their power. On the other hand, the challenge is supposed to inherently grant some power to the front-line workers for 100 days. It would be interesting to investigate whether the shape of the triangle or the predictive recommendations alter significantly were the study conducted over a longer duration (i.e., 150 days, 200, or more). Furthermore, existing beliefs that increased resources or policy changes are the main drivers of a systems change should be reconsidered. Perhaps there is an overreliance among street-level workers on administrators to change policy among other existing options for systems change.

Surprisingly, popular dialogue often revolves around changes in resources, such as increased funding. Yet, we find that the networking strategies and practices are more *commonly* utilized or implemented to achieve successful systems change. This could be due to the lack of control workers have over implementing changes in their resources, which also speaks to the high confidence that can be seen among power and resources in that those can rarely be changed.

However, some of these findings were no surprise to RE!NSTITUTE™. One worker summarized a common theme among the sentiments of workers who participate in the 100-Day Challenge™ saying,

> *We heard [from workers] 'One of the number one things that emerged from the challenge for us was the fact that I met a lot of people [other workers] who I hadn't met before and it made it more fluid for me to be able to do my job. Because it was useful, I will maintain those relationships moving forward'…Outreach staff are actually sitting down now with housing focused case managers. Those kinds of relationships get built together because that's the nature of putting together a multidisciplinary team. And the practice becomes things like better warm handoffs.*

There is also a link between relationship building and practice because when workers in different teams establish a connection, such as a warm handoff, an alternative workflow can be established to streamline efforts and maximize productivity for all parties (i.e., all agencies' success outcomes improve). The relationships they build can then evolve into varying newer and better practices.

When we consider the increased success rates in Northern Illinois Country and Desert City, presumably once relationships and new practices are established, the change points we found invite further inquiry. Increased rates of housing people who experience homelessness plateaued. Conversely, we feel that that the same can go for hiring *additional* workers to help spread the workload. If more workers were available for the team to implement these strategies, then increased success could surge and continue even higher, and absenteeism could be better accommodated for, too. We have, in this way, quantified the impact of sudden shifts in business operations by identifying marks of substantial changes in the rates at which people are relocated over time.

Some may attribute these stunted periods to be due to lags in reporting from workers. We find this to be the case when surges directly follow such lags, but the data does not show this to be the case in all instances. Lags without a corresponding

and subsequent surge implies that a major change was made (positive or negative), which includes workers missing work, which implies a necessity for the sustainability of workers over time to satisfy maintained success. By spotlighting the changepoints over time, we can see various times when growth in the success rates of housing people who experience homelessness get altered substantially. We can speculate that workers miss work due to many reasons, such as being overworked and understaffed (i.e., one person performing the tasks of many job titles), which could potentially trigger such change points. In future runs of these studies, we can probe into these possible causal factors. For example, as the last change point on Fig 5 for Northern Illinois shows, absenteeism could have prevented around 15 people from being relocated. We also believe that it indicates the opposite, too, that increasing the number of workers could lead to a substantial improvement in success rates. Calling attention to the retention of workers playing a key factor in sustaining improved outcomes reinforces a major finding of Greenfield and Pope [16]. Without worker retention, especially veteran leaders, then it stands to reason the continued success would suffer. We intend to delve deeper into this hypothesis in our future research.

While systems change projects can bring about significant benefits, it is crucial to consider the potential for unequal distribution of these benefits. Projects must be designed with equity in mind to ensure that all segments of the population can access and benefit from the changes. This is an interesting avenue to explore as we collect data over future years. Additionally, continuous evaluation and adaptation of these projects are necessary to address emerging challenges and ensure sustained positive outcomes.

### Theoretical implications

The strategies considered in this study reinforce Lipsky's [6] theories on street-level bureaucracy. The significant role that front-line workers play in enacting changes in practices and relationships aligns with Lipsky's assertion that these workers are crucial to policy execution. Their ability to modify everyday routines and broaden their networks is essential for overcoming bureaucratic hinderances to enact systemic change. The transformative change in mental models, as indicated by our findings, often depends on the adaptations made by street-level bureaucrats. Their buy-in and altered mental models are vital for enduring systems change, reinforcing the importance of supporting and empowering these workers.

Pattern-based rules can have the effect of limiting the discretion available to frontline public service workers, such as social workers and case managers. In theory, this can serve to safeguard against any form of bias or unfair decision-making. However, this restriction on discretion can also have the unintended consequence of stripping these workers of their ability to utilize professional judgment or modify their approaches to accommodate the specific and unique needs of the clients they serve. As noted by Lipsky [5] in 1980, an excessive reduction in discretion by inflexible systems can transform workers into simple implementers of policy, rather than enhancing their role as client advocates.

Sustained success in transformative change, as evidenced by the 100-Day Challenge™, is contingent upon the stability and sustainability of workers as street-level bureaucrats. These workers' practical execution and continuous presence are crucial to maintaining progress and achieving long-term systemic improvements. Implementing Lipsky's work into the study of systems change provides robust evidence of how equitable systems change can be achieved and sustained through the diligent efforts of these critical actors in the implementation landscape.

Our findings reinforce ethnographic scholars who study Lipsky's [6] work, such as Evelyn Brodkin [26,27], who argue that performance measurement instruments (PMIs) do little to productively measure successes among social service work, and the omnipresent discretion among workers is, in fact, at odds with such assessments. On this, Lipsky [6] specifies that workers' discretion inevitably makes street-level bureaucrats "virtual policy creators" as they reshape how formal policy is enacted in their daily work routines (p13). Workers can creatively negotiate their own creative implementation of the formal rules that oversee them. They exhibit an ability to satisfy the formal demands of their job via informal means, as they fit. While intuition may suggest that many administrators focus on punitive oversight to manage low-level workers who cut corners and fail their clients, it is vital to recognize that workers may have valuable insights in streamlining efforts to successfully reach programmatic goals.

Administrators who are willing to participate in collaboration with their front-line workers, such as in RE!NSTITUTE™'s 100-Day Challenge™, strategically encourage the valuable insights from workers to maximize all efforts for success. Veteran social service workers are especially good at creatively enacting their own means to effectively "fit" their clients into existing programmatic criteria [55] and to maximize the services available to their clients by overcoming the bureaucratic "red-tape" that prevents them from accessing aid [4].

We echo a growing call to action that contends that quantitative research be better informed by existing qualitative literature on topics of social services and homelessness. It is rare to find quantitative research on Lipsky's work on street-level bureaucracy because considering contextual factors is traditionally more the focus of qualitative research. One possible explanation is the rise of performance measures and performance measurement instruments (PMIs) that have taken the spotlight [56], despite their dubious reliability and limited success beyond their adoption by participating agencies [57,58]. However, quantitative research that is informed by qualitative literature is necessary to further our collective knowledge and inform better policy. In the *Social Service Review*, Mathys et al. [59] draw on 143 quantitative surveys to supplement their qualitative data. Their work serves as a bridge between both research methodologies. Like other qualitative researchers, Mathys et al [56] find that instruments that are designed to measure service outcomes are often unable to capture successes or failures among government-imposed performance measurement instruments (PMIs), saying:

*In addition to factors regarding the organizational culture that could account for variations among social workers, the instrument itself may be ill equipped to capture sufficiently the complexity underlying some of the case profiles. (p15-16)*

We reinforce these findings by spotlighting surges in success during the 100-Day Challenge™, when front-line workers are given increased discretionary power over their employment outcomes. Our study incorporates quantitative data to complement the findings of prior qualitative research, ever since Lipsky's [5] foundational theories in 1980. We show that by considering contextual variables, such as the views of street-level bureaucrats, can contribute to literature on best practices and better inform social policy.

Research on systems change also emphasizes that administrators be open-minded to the views of ground-level workers. For instance Kania et al. [13] says:

*Any organization's ability to create change externally is constrained by its own internal policies, practices, and resources, its relationships and power imbalances, and the tacit assumptions of its board and staff… In addition, funders cannot support efforts that run counter to their own mental models. The implications of this are daunting. To fully embrace systems change, funders must be prepared to see how their own ways of thinking and acting must change as well. (p5)*

The principles spotlighted by Kania et al. [13] underline the significance of reflection, flexibility, and holistic thinking in systems change efforts. They also say, "Unless funders and grantees can learn to work at this third level [mental model], changes in the other two levels will, at best, be temporary or incomplete [13] (p 8). This reinforces the need for administration's willingness to consider the perspective of front-line workers as an important part in improving success, as Lipsky [5] first emphasized.

We've shown how workers streamline their workload by changing how they network and practice the demands of their job. Like Lipsky [6], we want to emphasize that instead of punitive standardization of their employment outcomes, social service workers (and potentially any public service workers) thrive best when given the opportunity to maximize their performance by working in collaboration with administrators.

**Practical implications, limitations, recommendations, and RE!NSTITUTE™'s operational insights**

Typically, most social service agencies struggle to measure success via pretest and posttest or conduct follow-up interviews, given their overworked employees and limited resources. It is often just not feasible to do much more. While this is true for workers in social services, we believe our contribution goes well beyond traditional measurement among social service work. Our analysis shows that systems change is possible with the recipe we have laid out, along with a measurement for maintained success and comparative strategy implementation. We understand that given the right conditions does not guarantee success. We also would like to be able to compare success rates across many geographic locations, if given the opportunity. One limitation is that success measurements were only available for Northern Illinois and Desert City in this study. Additionally, although we can see the varying confidence levels represented by the vertical height of each triangle in Fig 2, we cannot establish a threshold for how substantial (i.e., confident or not) the confidence level is. Future research should seek to show such a threshold to show if an item is "substantially confident" or not.

Moreover, this study was time consuming and not easily replicated. Had this study been replicated many times (or 0.1613 changed in replications) the confidence could potentially show to be higher. However, we do represent the confidence in the vertical representation (i.e., how tall) of each triangle in Fig 2. The confidence for changes in policy, resources, power, and mental changes were higher for a systems change. Perhaps, this is due to front-line workers' lack of ability to change policy, resources, power (or the true power despite their supposed freedoms during the 100-Day Challenge™). Most, if not all, workers do not have the ability to impact resource flows, for example.

We can assume that continued implementation of the challenge beyond 100 days would result in continued success. Since we see steady increased success throughout the 100-Day Challenge™, there is no reason to think that this success would decline in the future with continued performance. With the same line of thinking, we can speculate that if additional workers were to be hired, then we can expect the successes to surge, since we find that successes plateaued when participating workers missed work.

In general, there are multiple reasons why the rate of housing homeless clients may change over the course of a 100-Day Challenge™. One primary reason is the availability of new housing units. When more housing units become available during the challenge, it increases the capacity to house individuals and families, thereby affecting the overall rate of housing placements. Another significant factor is the introduction of policy changes or innovations by the team that expedite the housing process. These can include strategies like cross-system case conferencing, relieving documentation burdens, improving coordinated entry, or increasing housing navigation staff. Such changes streamline the process individuals must go through to obtain housing, thus improving efficiency and speed of placement. Additionally, the team might implement new innovations or develop relationships to better identify those experiencing homelessness. Doing so means that more individuals are recognized and added to the list of those seeking housing and services. An enhanced identification process ensures that more people are counted and assisted, which can increase the housing rate. Finally, data management, including cleaning up data, can significantly affect housing rates. Teams may correct mistakes or add new information to the database, which can reflect as an increase or decrease in the reported number of people connected to housing. Accurate and up-to-date data is crucial in understanding and improving housing outcomes.

There could also be many potential reasons behind changes in the two cases highlighted in Fig 5a and 5b. For instance, in Desert City, The 100-Day Challenge™ provided the opportunity for the community to implement several action items from the Desert City Plan to End Youth Homelessness [60], which is the first plan ever to end youth homelessness in the region which was released just weeks before the challenge began.

New cross-system collaboration between youth-serving systems and agencies may have helped the Team improve identification of systems or involved youth who are most vulnerable to exiting into homelessness and hold frequent meetings to case conference the specific needs of youth at risk of homelessness. In particular, the juvenile justice and child welfare systems were able to identify a significant crossover in population served. Through the 100-Day Challenge™,

these systems work together to leverage existing resources and broaden homelessness prevention efforts to serve youth with dual system involvement.

Desert City compiled a collaborative, cross-system list of youth at risk of homelessness and tracked their experience through the systems in which they are involved. Team members including representatives from Department of Family Services (DFS), the Department of Juvenile Justice (DJJS), the Nevada Youth Parole Bureau, and numerous housing and homeless service providers used this list to conduct case conferencing for youth identified as exiting a public system and at risk of homelessness.

By sharing information on resources and housing options available for youth experiencing homelessness, the community's housing and homeless service providers could better serve this population. They built an online tracking tool for housing options available to youth experiencing homelessness, including landlords willing to work with this population. This meant that team members could easily navigate to the best possible housing option for a client based on their needs and circumstances.

Frequent communication across systems also helped partners realize that they were serving many of the same youth and encouraged them to reach across the table for assistance in referring youth to appropriate services.

A collaborative team of social service workers engaged in weekly calls that brought together representatives from these diverse systems, which helped hold each other accountable to their programmatic goals and foster stronger relationships between agencies. The cross-system relationships built and strengthened during the 100-Day Challenge™ resulted in better identification of youth and access to more community resources.

The Northern Illinois 100-Day Challenge™ team employed a multi-faceted approach to improve identification of unaccompanied youth experiencing homelessness. The Team leveraged the new relationships through the 100-Day Challenge™ work to improve outreach efforts within non-CoC systems, including schools, state agencies, employment, justice, and community groups. Northern Illinois supported these increased outreach efforts in non-CoC systems through the development of robust marketing materials, including flyers and posters, to connect with youth who had not yet been identified as experiencing homelessness and connect them to resources. As a result of these efforts, Northern Illinois saw the consistent addition of youth experiencing homelessness to the active by-name list, indicating that the increased outreach and identification efforts were working to reach new youth experiencing housing instability within local systems.

Finding that mental models is the single most significant in achieving transformative change is especially telling in the context of the 100-Day Challenge™. This finding involves workers shifting the perspective away from the idea of independent agencies being "siloed" or isolated from each other, usually due to competition over funding allocations. Once workers subscribed to the idea of working together across roles, implementing strategies such as warm hand-offs from one agency to another, they often reported to RE!NSTITUTE™ that they were viewing things from the perspective of the client rather than the criteria dictated by their agency's employment standards and goals. We found that power factors also dictated much of the other variables, which we attribute to how much agencies chose to enact the 100-Day Challenge™ by granting power to front-line workers. If workers were granted such power, then this often impacted how greatly mental buy-in could be satisfied for change to occur in their work practices, often resulting in substantial increased success. In short, we show that leadership needs to allow workers to behave differently from their traditional employment standards during the 100-Day Challenge™ for more change to happen, which directly impacts success. Additionally, if we can guarantee changes in power and resources during the 100-Day Challenge™, changes in policy are seen to result in half (50%) of the participating agencies. Afterwards, participating agencies added that their local collective policy implementation per their official Continuum of Care (i.e., their localized board overseeing funding from the United States Housing and Urban Development) was changed.

It may be apt to query whether the mental changes brought about through the 100-day challenge are meant to last. We note here that social change projects, even when implemented over short periods, can substantially influence mental changes, practices, and relationships that benefit the population [61,62] (Oudsoorrn 2021) (Turner 2023). These projects

often focus on enhancing social security, improving mental health, and fostering social relationships, which are crucial for societal well-being [63–65] (Daly 2013; Evens WN 2021; Tummers 2014). Social security policies, for instance, have a profound impact on mental health [66] (Ponka). Changes in social security benefits can influence mental health outcomes such as stress, anxiety, and depression. A systematic review [67] (Simpson)shows this relationship, emphasizing the importance of policy in shaping mental health outcomes. Short-term interventions, like creative arts therapy programs, have shown significant improvements in children's social behaviors and relationships, indicating that even brief, intensive programs can positively affect mental health and social skills [68]. Organizational change readiness is influenced by social relationships in the workplace [69] (Mcarthys). Strong social ties and organizational commitment can facilitate smoother transitions during change, highlighting the importance of fostering supportive environments in social change initiatives [70]. Public participation in infrastructure projects can lead to social benefits, although its development is slow. Engaging stakeholders in the planning and execution of projects can enhance social outcomes and address socio-economic disputes [71].

The construction and continuous strengthening of these projects can lead to sustainable benefits for the population, as they address immediate needs while laying the groundwork for long-term improvements. The benefits of these projects can be compared by examining their impact on mental health, social relationships, and sustainable practices. The benefits of social change projects, such as our 100-day challenge, can be compared by examining their impact on different population groups. For instance, smart mobility policies in the Netherlands showed differential benefits, with car users gaining more than students, raising questions about social equity in policy design [72].

Considering each geographic location that participates in the 100-Day Challenge™ has differing strategies and implementation rates that lead to variations in outcomes, we would like to also include their success information. For instance, Northern California (Sacramento), evidenced in Fig 6, to be unattached to most of the other counties, may be ahead or behind all the rest, which means that they could be models for other future projects to emulate in case the changes are due to strategies that improve success and are implementable and sustainable. The inclusion of their successes rates would be even more meaningful than what we have done here.

### Future research

Future research should focus on conducting follow-up with workers to see how they are conducting their work six months and one year after the 100-Day Challenge™, focusing efforts to measure if they are still utilizing the same strategies and having the same successes as during the challenge.

Additionally, success measures of housing people experiencing homelessness outside of Northern Illinois and Desert City have not been compared to this model, yet. We presume that adding measures of successful cases to weigh against the ingredients of a systems change may alter our perceptions of desirability. For instance, increased relationships or alternative practices may or may not necessarily bring about clients' successful transition into housing, just successful changes in ways systems run. While we feel this study could bring substantial change to inform social policy in the context of successful intervention of a 100-Day Challenge™, in general, weighing *detailed* success outcomes against what we have here would be ideal.

The structural (triangle), correlational (market basket) insights and change point estimates showcased here (the time switched statistic) generalize effortlessly to similar or even more nuanced settings. For example, as of this writing, we are monitoring data coming from the Department of Education and three local school systems in Connecticut to measure the impact of operational tweaks on homelessness programs with special emphasis on the student population. Preliminary results suggest a drop of around 8% in such student population. We are looking at the veteran population in Illinois and New York who may need help in filling out survey forms accurately to stop being moved around in and out of homeless shelters, a separate study in Oregon with sleep trailers, etc. These are instances of shifts in ways of working. These are ongoing studies and are examples of shifts in ways of working (that is, "practice"). We are beginning to notice near

simultaneous shifts in relationship among the workers and mental attitudes (but less so in power dynamics and resources) just as the recommendation networks with "Practice LHS" suggests). Generalizability is, thus, evidenced in a potentially nonstationary sense: since the ingredients here are macro properties, so to speak, such as power dynamics or resources which do not fluctuate overnight, recommendation shifts in tendencies are likely to remain valid as time moves. The choice of states, we stress, is not guided necessarily by political or demographic dictates, rather by whichever state authorities partner with the RE!NSTITUTE™ [29] with sound funding and legal agreements. Moving forward, due to the success on display and the quantitative justification backing it, we sense many of the states will be welcoming of these collaborations.

## Supporting information

**S1 File. Appendix A, Appendix B, and Appendix C to go through workers' first-hand experiences and personal accounts along with mathematical subtleties.** Tessellations [42] and other geometric insights are used to stress hierarchical subtleties.
(DOCX)

## Acknowledgments

We want to thank RE!NSTITUTE™ [29] for allowing us access to the data and for helping to build a great working relationship.

## Author contributions

**Conceptualization:** Curtis Smith, Moinak Bhaduri.

**Data curation:** Moinak Bhaduri.

**Formal analysis:** Curtis Smith, Moinak Bhaduri.

**Funding acquisition:** Curtis Smith, Moinak Bhaduri.

**Methodology:** Moinak Bhaduri.

**Supervision:** Curtis Smith.

**Visualization:** Moinak Bhaduri.

**Writing – original draft:** Curtis Smith, Moinak Bhaduri.

**Writing – review & editing:** Curtis Smith, Moinak Bhaduri.

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
