## [Decision Letter · Decision Letter 0]

4 May 2025

PONE-D-24-43337A recipe for systems change: Predictive modeling and street-level bureaucracy among homeless services.PLOS ONE

Dear Dr. Smith,

Thank you for submitting your manuscript to PLOS ONE. After careful consideration, we feel that it has merit but does not fully meet PLOS ONE’s publication criteria as it currently stands. Therefore, we invite you to submit a revised version of the manuscript that addresses the points raised during the review process.

We look forward to receiving your revised manuscript.

Kind regards,

Mohammad Salah Hassan, Ph.D

Academic Editor

PLOS ONE

Journal Requirements:

[NO].

3. Thank you for uploading your study's underlying data set. Unfortunately, the repository you have noted in your Data Availability statement does not qualify as an acceptable data repository according to PLOS's standards.

[This work was supported, in part, by the Research Enhancement Grant awarded to the second author by the American Mathematical Society and the Simons Foundation for the 2024-2027 cycle.]

[The author(s) received no specific funding for this work.]

Additional Editor Comments:

Dear Authors,

Thank you for submitting your manuscript, "A Recipe for Systems Change: Predictive Modeling and Street-Level Bureaucracy Among Homeless Services." Your efforts to explore the intersection of frontline discretion and predictive modeling in the context of homelessness services is both timely and intellectually ambitious. The paper presents a valuable attempt to combine street-level bureaucracy theory with data-driven analysis and dashboard applications, which is a welcome contribution to both public policy and applied social science.

That said, while both reviewers have recommended minor revisions, I believe the manuscript would benefit from more substantial work before it can be considered for publication. In particular, the narrative structure and methodological presentation require refinement to enhance clarity and accessibility for a broader readership.

Some sections, especially those covering technical modeling and visual representations, are overly dense or abstract. Readers unfamiliar with the mathematical frameworks may find it difficult to follow, and I encourage you to streamline explanations while maintaining scientific rigor. Consider moving detailed formulaic content or extended metaphors (e.g., shopping carts and recipes) into a supplementary file, and focus in the main text on why the methodology matters in practical terms for systems change and policy implementation.

Tone and language also need attention. Phrases like "throwing money at the problem" and comparisons to "a competent chef" detract from the otherwise academic quality of the manuscript. Please revise for more consistent scholarly language throughout.

Additionally, the integration of figures and tables was uneven in the version reviewed. Please ensure all visuals are present, clearly labeled, and thoroughly explained in the text. Some figures, especially those central to the triangle model and dashboard logic, need stronger captions and contextual grounding.

I also encourage you to strengthen the literature review component. While relevant sources are cited, particularly in relation to Lipsky’s theory, performance measurement tools, and the RE!NSTITUTE framework, these references are scattered throughout the introduction and discussion without a cohesive structure. Consider consolidating and deepening your engagement with prior research in a dedicated section or a more organized narrative format. This will help clarify the theoretical grounding of your work and better situate your contribution within existing scholarly conversations.

Beyond presentation, I would like to see a fuller discussion of limitations, particularly regarding sample size, generalizability, and potential biases in dashboard design or interpretation. Clarifying the nature of the data (e.g., ethics, consent, and de-identification) would also strengthen transparency.

Finally, please respond in detail to the reviewers’ comments. While they found the work promising, several suggestions, such as clarifying theoretical framing and improving flow—align with the concerns I’ve raised here.

This is a promising paper, and with thoughtful revision, it could make a strong contribution. I invite you to revise the manuscript accordingly and submit a point-by-point response addressing all reviewer and editor feedback.

Sincerely,

Mohammed Salah, PhD

Academic Editor

PLOS ONE

Reviewers' comments:

Reviewer's Responses to Questions

**Comments to the Author**

1. Is the manuscript technically sound, and do the data support the conclusions?

Reviewer #1: Yes

Reviewer #2: Partly

2. Has the statistical analysis been performed appropriately and rigorously? 

Reviewer #1: Yes

Reviewer #2: Yes

3. Have the authors made all data underlying the findings in their manuscript fully available?

Reviewer #1: Yes

Reviewer #2: No

4. Is the manuscript presented in an intelligible fashion and written in standard English?

Reviewer #1: Yes

Reviewer #2: Yes

5. Review Comments to the Author

Reviewer #1: Thank you for the invitation to read this interesting and important paper. I find it very well written and coherent. THe theoretical framework and methods are sound. There are some issues that would be possible enhance the author´s argument:

1. some more theory on systems change than Kania et al should be included

2. some more reflections on what is the research gap that the article is addressing should be more explicit and grounded in the state-of-art.

3. the discussion lacks a link between results and scientific/theoretical literature.

Reviewer #2: The work is very ambitious to interpret the results derived from the application of both the MBA and network models, as it reflects only associations and not causalities, which reduces the possibility of reliable certainty in decision-making processes.

Although it is showed that the geographic issue will be addressed, the text needs to contextualize how projects of this magnitude in such a short period are consistent with the mental changes, practices, and relationships that make a difference, how their construction and continuous strengthening benefits the population, and how the benefits are compared.

In the analysis, they address the importance of networks integrated into legal assistance services for homeless people, medical care, and employment services, for example. This could establish other patterns.

Another issue is replicability and discussing case studies with heterogeneous territorial spaces and social processes from a quantitative perspective, leaving aside the qualitative, which may have very specific implications for the area.

MBA algorithms can struggle with large data sets. How did you control this in your model?

The decision on rules for weakness and strength is not explicit in the text.

With the model used, occurring processes is known, the MBA can act as a “decision aid” by revealing non-obvious patterns, including how these relate to beneficiary behaviors and retention. Or, what are the tools to detect other relationships, such as those related to health? For example, a frontline worker might not realize how frequently substance abuse treatment is related to housing retention unless the data shows it.

Is service fragmentation reduced? Will data prevail over local discretion? Could pattern-based rules lead to rigid or unfair decision-making?

6. PLOS authors have the option to publish the peer review history of their article (what does this mean? ). If published, this will include your full peer review and any attached files.

**Do you want your identity to be public for this peer review?** For information about this choice, including consent withdrawal, please see our Privacy Policy .

Reviewer #1: No

Reviewer #2: No

---

## [Author Response · Author response to Decision Letter 1]

10 Jun 2025

Response to editor and referee comments. PLEASE SEE THE ATTACHED FILE WHERE WE HAVE DETAILED RESPONSES TO EACH CONCERN OF THE EDITOR AND ALL THE REVIEWERS.

Dear Authors,

Comment 1: Thank you for submitting your manuscript, "A Recipe for Systems Change: Predictive Modeling and Street-Level Bureaucracy Among Homeless Services." Your efforts to explore the intersection of frontline discretion and predictive modeling in the context of homelessness services is both timely and intellectually ambitious. The paper presents a valuable attempt to combine street-level bureaucracy theory with data-driven analysis and dashboard applications, which is a welcome contribution to both public policy and applied social science.

Response: We thank the editor for the thoughtful comments and praise. Yes, the above is an apt synopsis of our article. We welcome the feedback, and we feel that the comments provided help improve the manuscript considerably. We thank you and the reviewers for the thoughtful feedback.

We have attached a file with track changes, and we included notes in the document that reference each concern listed here. In doing so, we hope this helps in checking that we have addressed every concern.

Concern 2: That said, while both reviewers have recommended minor revisions, I believe the manuscript would benefit from more substantial work before it can be considered for publication. In particular, the narrative structure and methodological presentation require refinement to enhance clarity and accessibility for a broader readership.

Response: Great feedback. As can be seen in the revised copy, particularly in the file with track-changes, we have heavily revised the order of the introduction and key sections of the manuscript by paying close attention to narrative flow and clarity. For example, we added section titles, such as “Systems change: A brief synopsis” and “Michael Lipsky’s “street-level bureaucracy” to adequately delineate the topics for easier readability.

Concern 3: Some sections, especially those covering technical modeling and visual representations, are overly dense or abstract. Readers unfamiliar with the mathematical frameworks may find it difficult to follow, and I encourage you to streamline explanations while maintaining scientific rigor. Consider moving detailed formulaic content or extended metaphors (e.g., shopping carts and recipes) into a supplementary file, and focus in the main text on why the methodology matters in practical terms for systems change and policy implementation.

Response: Technical discussions have been simplified further by adding explanatory details (shown in the responses below) both in text and in the figure captions. We have separated the mathematical descriptions from the main text and added the following in place:

We point readers interested in mathematical technicalities to a crucial appendix (Appendix C) we have designed for this purpose. In summary, we track the times at which homeless people get relocated, construct a number (detailed in appendix C) that serves as the rate of relocations, and used this number to track the earliest instance of a changed rate over the course of the 100 days. Once this earliest instant is found, we begin the algorithm afresh to detect the next possible point(s) of change. Patterns in these detected “change” times hold particular relevance, especially in tracking how time-bound relocation exercises such as the 100-day challenges progress, with and without external (such as governmental) supervision. For instance, with time-bound exercises, a stronger concentration of such change times may be present towards the end of the fixed period which could reflect workers’ upped urgencies. One may compare the efficacy of two time-bound attempts with how similar their change points were, or compare one time-bound study to a general relocation attempt. These “change” times also enable us to measure the impact of a shift. We can contrast the forecasts from the pre-change piece (had there been no change) to the observed relocation numbers from the actual, post-change piece. And possibly ascribe the gap - using field of knowledge - to preventable causes such as absenteeism. The strategy described above helps pinpoint where changes occurred, that is, deliver solid definitions of the pre- and post-change pieces, crucial for these gap measurements. We look at two specific examples below to elaborate these matters.

Concern 4: Tone and language also need attention. Phrases like "throwing money at the problem" and comparisons to "a competent chef" detract from the otherwise academic quality of the manuscript. Please revise for more consistent scholarly language throughout.

Response: We agree that the language is inflammatory, and the feedback is welcomed. We revised with more academic language for the examples provided here and throughout the paper.

Concern 5: Additionally, the integration of figures and tables was uneven in the version reviewed. Please ensure all visuals are present, clearly labeled, and thoroughly explained in the text. Some figures, especially those central to the triangle model and dashboard logic, need stronger captions and contextual grounding.

Response: We apologize for the scant captions on these figures. The captions have now been enlarged to accommodate details as follows:

Figure 1: Theoretical contributions from Kania et al. (2018).

has been changed to

Figure 1: Theoretical contributions from Kania et al. (2018). A conceptual triangle is hypothesized which brings out educated guesses on how common certain kinds of changes are likely to occur in relocation exercises. The wider the base, the easier to bring about that change. We notice, therefore, how structural changes are guessed to be easier to bring about than mental changes. We notice, however, two points. First, in this conceptual map, there is no backing from observed data. Second, the cosmetics do not reveal insights: that is, the base or the altitude of the component shapes do not hold special significance. Both these points are taken up in figure 3, a realized version of this conceptual triangle through statistical metrics.

Previous version of Figure 2: Frequency of change by theme according to data from RE!NSTITUTE™, based on a theoretical model for “Systems Change” from Kania et al. (2018) using mathematical concepts from Bhaduri (2023).

has been changed to

Figure 2: Frequency of change by theme according to data from RE!NSTITUTE™, based on a theoretical model for “Systems Change” from Kania et al. (2018) using mathematical concepts from Bhaduri (2023). The heights of the towers describe the rate at which changes in the corresponding types were witnessed to occur. For instance, of all the 100-day challenges studied, we found about 55% of times changes in “relationship” were observed, either changes in “relationship” only, or in “relationship” along with those in other types.

Previous version of Figure 3: Legitimacy of theoretical model for “Systems Change” from Kania et al. (2018) by theme according to data from RE! RE!NSTITUTE™ and mathematical concepts from Bhaduri (2023).

has been changed to

Figure 3: Legitimacy of theoretical model for “Systems Change” from Kania et al. (2018) by theme according to data from RE! RE!NSTITUTE™ and mathematical concepts from Bhaduri (2023). We interpret this as a realized version on the theoretical triangle shown in figure 1. The shapes and the sizes of the components have been calculated using data from the 100-day challenges. Unlike Figure 1, the bases and the altitudes of the shapes here do convey statistical information. The bases represent how common changes in a certain category are (the wider the base, the more common the change) while the altitudes represent the inverse of the length of the confidence interval for this statistic (the taller the altitude, the more sure we are in the value of the base). This observed empirical triangle (Figure 3), therefore, confirms, in the process, the general shape of the theoretical triangle (Figure 1) in that the bases for “practices”, “policy”, and “resources”, collectively (that is, “structural”) cover the widest ground and the bases subsequently taper down on relational and mental changes.

Previous version of Figure 4: A typical map of likely connections that lead to changes in mental structures. The more likely connections are shown in deeper red. We invite readers to explore other possibilities (for instance, those connections that lead to non-mental targets) on our interactive dashboard.

has been modified to

Figure 4: A typical map of likely connections that lead to changes in mental structures. The more likely connections are shown in deeper red. We invite readers to explore other possibilities (for instance, those connections that lead to non-mental targets) on our interactive dashboard. One implements these rules in an input-output sense. If changes in the input details - shown by the blue arrows - are promised, one is given the rate at which changes in the output detail - shown by the red arrow - accompany. The higher this rate, that is, the more (correlationally) automatic the input-output connections are, the darker the shade of the red node representing the rule. For instance, rule 12 says if one designs a relocation study that generates changes in “relationship”, “practice”, “power”, changes in “mental” structures will be promised with a reasonably high chance (50% if one hovers on the dashboard).

Previous version of Figure 5: Epochs of stable relocation rates are shown on either side of the red vertical separators for (a) Northern Illinois and (b) Desert City.

has been changed to

Figure 5: Epochs of stable relocation rates are shown on either side of the red vertical separators for (a) Northern Illinois and (b) Desert City. Each vertical separator represents an estimated change-point, that is, a point in time over the course of the 100 days during which the rate of relocating homeless people improved or deteriorated substantially. For instance, had the change-point around day 80 (=400/5) not been there for Northern Illinois, the total number of people relocated could have been 85 instead of 80. Crowding of change points towards the end of the experiment is observed for nearly all regions.

Concern 6: I also encourage you to strengthen the literature review component. While relevant sources are cited, particularly in relation to Lipsky’s theory, performance measurement tools, and the RE!NSTITUTE framework, these references are scattered throughout the introduction and discussion without a cohesive structure. Consider consolidating and deepening your engagement with prior research in a dedicated section or a more organized narrative format. This will help clarify the theoretical grounding of your work and better situate your contribution within existing scholarly conversations.

Response: We have revised to lengthen and substantiate the literature review portions of the manuscript, and feel gratitude for this feedback. We feel a sense of gratitude for the opportunity to find more current articles on the topic from the past five to seven years, including one from 2025, which spotlights the relevance of the topic. We feel that this revision helped solidify the topic’s rising importance. You can view these changes at the beginning and end of the section titled Systems change: A brief synopsis and in Michael Lipsky’s “street-level bureaucracy.”

We revised the entire introduction and literature review for clarity and also feel this aided in the revision process of your Concern #2. We feel that the message of the manuscript is much stronger now. Thank you.

Concern 7: Beyond presentation, I would like to see a fuller discussion of limitations, particularly regarding sample size, generalizability, and potential biases in dashboard design or interpretation. Clarifying the nature of the data (e.g., ethics, consent, and de-identification) would also strengthen transparency.

Response: We thank the editor for raising this. Limitations have been clarified in the revision in the following way:

“While systems change projects can bring about significant benefits, it is crucial to consider the potential for unequal distribution of these benefits. Projects must be designed with equity in mind to ensure that all segments of the population can access and benefit from the changes. This is an interesting avenue to explore as we collect data over future years. Additionally, continuous evaluation and adaptation of these projects are necessary to address emerging challenges and ensure sustained positive outcomes.”

Regarding sample size, we have added:

“These connections - we stress again - represent correlational links for the moment and need not be taken as causal pathways. In our previous works - Bhaduri (2024), Bhaduri (2025), Camp et al. (2024), Zahirodini and Bhaduri (2025) - we have explored this matter of causality in analogous social settings using similar network structures - Markov random fields - to showcase how gender of respondents and political leanings causally sway their opinion towards current business practices, how students from diverse backgrounds express their sentiments towards the living conditions of migrants at the US-Mexico border, etc. The estimation of these random fields, however, depends on the delicate balance of discerning false positives and false negative edges through regression techniques such as lasso which are data hungry algorithms. In view of the fact that this is the first of a long sequence of ongoing 100-day challenges, data, though of high quality, are relatively scarce, and we have refrained from querying causal effects. In time, as similar high quality data become available, we can deploy the Markov random fields much along the lines of the works cited earlier to explore more interesting causal pathways.”

Regarding generalizability, we have added (in addition to our reply to referee 2's observation on causality - another generalization concern - and the crash example we added now):

The structural (triangle), correlational (market basket) insights and change point estimates showcased here (the time switched statistic) generalize effortlessly to similar or even more nuanced settings. For example, as of this writing, we are monitoring data coming from the Department of Education and three local school systems in Connecticut to measure the impact of operational tweaks on homelessness programs with special emphasis on the student population. Preliminary results suggest a drop of around 8% in such student population. We are looking at the veteran population in Illinois and New York who may need help in filling out survey forms accurately to stop being moved around in and out of homeless shelters, a separate study in Oregon with sleep trailers, etc. These are instances of shifts in ways of working. These are ongoing studies and are examples of shifts in ways of working (that is, “practice”). We are beginning to notice near simultaneous shifts in relationship among the workers and mental attitudes (but less so in power dynamics and resources) just as the recommendation networks with “Practice LHS” suggests). Generalizability is, thus, evidenced in a potentially nonstationary sense: since the ingredients here are macro properties, so to speak, such as power dynamics or resources which do not fluctuate overnight, recommendation shifts in tendencies are likely to remain valid as time moves. The choice of states, we stress, is not guided necessarily by political or demographic dictates, rather by whichever state authorities partner with the ReInstitute with sound funding and legal agreements. Moving forward, due to the success on display and the quantitative justification backing it, we sense many of the states will be welcoming of these collaborations.

Regarding potential biases in dashboard design (the below) and interpretations (the above on causal connections), we have added:

“Network objects are underexplored in the social sciences. Consequently, the dashboard, which brings out specific kinds of network structures, while extremely insightful, must be interpreted with some caution, in case one is unfamiliar with non-Euclidean mathematics (we direct such readers to

---

## [Decision Letter · Decision Letter 1]

8 Jul 2025

A recipe for systems change: Predictive modeling and street-level bureaucracy among homeless services.

PONE-D-24-43337R1

Dear Dr. Authors,

We’re pleased to inform you that your manuscript has been judged scientifically suitable for publication and will be formally accepted for publication once it meets all outstanding technical requirements.

Kind regards,

Mohammad Salah Hassan, Ph.D

Academic Editor

PLOS ONE

Additional Editor Comments (optional):

Thank you for addressing all the issues

Reviewers' comments:

Reviewer's Responses to Questions

**Comments to the Author**

1. If the authors have adequately addressed your comments raised in a previous round of review and you feel that this manuscript is now acceptable for publication, you may indicate that here to bypass the “Comments to the Author” section, enter your conflict of interest statement in the “Confidential to Editor” section, and submit your "Accept" recommendation.

Reviewer #1: All comments have been addressed

Reviewer #2: All comments have been addressed

2. Is the manuscript technically sound, and do the data support the conclusions?

Reviewer #1: Yes

Reviewer #2: Yes

3. Has the statistical analysis been performed appropriately and rigorously? 

Reviewer #1: Yes

Reviewer #2: Yes

4. Have the authors made all data underlying the findings in their manuscript fully available?

Reviewer #1: Yes

Reviewer #2: Yes

5. Is the manuscript presented in an intelligible fashion and written in standard English?

Reviewer #1: Yes

Reviewer #2: Yes

6. Review Comments to the Author

Reviewer #1: The authors have addressed all my concerns an this has improved the paper. Thus I I support a publication

Reviewer #2: (No Response)

7. PLOS authors have the option to publish the peer review history of their article (what does this mean? ). If published, this will include your full peer review and any attached files.

**Do you want your identity to be public for this peer review?** For information about this choice, including consent withdrawal, please see our Privacy Policy .

Reviewer #1: **Yes: ** Ottar Ness

Reviewer #2: No

---

## [Editor Report · Acceptance letter]

PONE-D-24-43337R1

PLOS ONE

Dear Dr. Smith,

I'm pleased to inform you that your manuscript has been deemed suitable for publication in PLOS ONE. Congratulations! Your manuscript is now being handed over to our production team.

Kind regards,

on behalf of

Dr. Mohammad Salah Hassan

Academic Editor

PLOS ONE